# Association Pattern-aware Fusion for Biological Entity Relationship Prediction

**Lingxiang Jia**[1] **Yuchen Ying**[1] **Zunlei Feng**[1,2]* **Zipeng Zhong**[1] **Shaolun Yao**[1]
**Jiacong Hu**[1] **Mingjiang Duan**[1] **Xingen Wang**[1,3] **Jie Song**[1] **Mingli Song**[1,2]

[1]State Key Laboratory of Blockchain and Data Security, Zhejiang University
[2]Hangzhou High-Tech Zone (Binjiang) Institute of Blockchain and Data Security
[3]Bangsheng Technology Co, Ltd.

## Abstract

Deep learning-based methods significantly advance the exploration of associations among triple-wise biological entities (e.g., drug-target protein-adverse reaction), thereby facilitating drug discovery and safeguarding human health. However, existing researches only focus on entity-centric information mapping and aggregation, neglecting the crucial role of potential association patterns among different entities. To address the above limitation, we propose a novel association pattern-aware fusion method for biological entity relationship prediction, which effectively integrates the related association pattern information into entity representation learning. Additionally, to enhance the missing information of the low-order message passing, we devise a bind-relation module that considers the strong bind of low-order entity associations. Extensive experiments conducted on three biological datasets quantitatively demonstrate that the proposed method achieves about 4%-23% hit@1 improvements compared with state-of-the-art baselines. Furthermore, the interpretability of association patterns is elucidated in detail, thus revealing the intrinsic biological mechanisms and promoting it to be deployed in real-world scenarios. Our data and code are available at https://github.com/hry98kki/PatternBERP.

## 1 Introduction

Exploring potential associations among triple-wise biological entities (e.g., drug-target protein-adverse reaction) [1–6] holds significant implications for elucidating underlying biological mechanisms and advancing personalized therapies [7–9], thus promoting pharmaceutical innovation and ensuring human health. Recent deep learning-based methods have propelled auxiliary prediction tasks concerning biological entity relationship, with most focusing on binary associations (e.g., drug-target protein), while only a few methods offer insights for more complex triple-wise associations. Existing solutions for the association prediction task can be broadly categorized into three types: (1) non-graph methods [10–12]; (2) graph-based methods [13–16]; (3) hypergraph-based methods [17–22].

As illustrated in Figure 1, non-graph methods typically concatenate the features of different entities, which are independently mapped by their respective entity encoders, to serve as representations. Graph-based methods adopt nodes and edges of the graph to represent entities and their relationships, and leverage the graph structure for feature propagation, thereby achieving effective representation learning of entity nodes. Similar yet distinct, hypergraph-based methods employ the hypergraph structure to obtain entity representations using the complex feature aggregation strategy. However, none of the aforementioned methods consider the significance of path patterns in the graph structure, which contain a vast amount of crucial information including hidden context and co-occurrence.

---

*Correspondence to zunleifeng@zju.edu.cn

38th Conference on Neural Information Processing Systems (NeurIPS 2024).

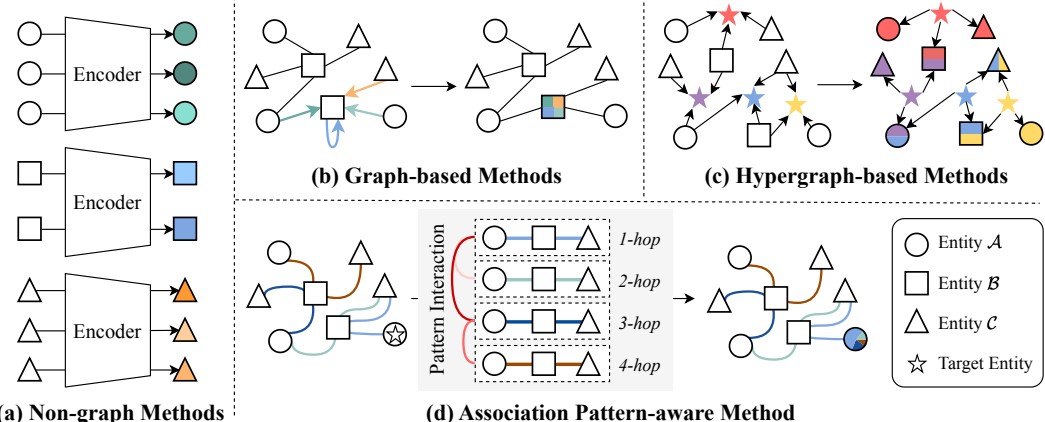

Figure 1: Comparisons of feature update strategy among non-graph methods, graph-based methods, hypergraph-based methods, and the proposed association pattern-aware method. Unlike existing methods that map or aggregate node features, the proposed method mines and then fuses association patterns for each target entity node in the graph to enhance the model's representative ability.

To this end, we introduce a novel association pattern-aware message propagation strategy as shown in Figure 1(d). The module leverages the potential relationships, such as commonality and diversity, of association patterns as the rule for facilitating message passing among entity features, which can efficiently expand the potential for representing complex interactions within the perspective of both basic graph structure and intrinsic biological mechanisms. Specifically, the related patterns within the graph structure are sampled through the pre-defined distance relation for each entity node. The message passing is driven by the interaction of its assigned patterns, i.e., the feature of the entity node is subsequently updated through feature fusion using adaptive coefficients that capture pattern commonality, generated during the pattern interaction stage. This process is followed by the acquisition of potential common patterns with genuine biological significance for various entities.

In this paper, we propose a novel Association **Pattern**-aware Fusion method for **B**iological **E**ntity **R**elationship **P**rediction, namely *Pattern-BERP*. First, we devise an association pattern-aware strategy to solve the limitation caused by entity-centric feature mapping and aggregation. The strategy utilizes the association patterns related to each entity node within the graph to extract the common feature based on the attention mechanism for these patterns, thus expanding the ability to represent hidden complex interactions. In addition, to preserve the information interaction of different entities, a hypergraph-based block is incorporated with the association pattern-aware fusion module, thereby enhancing the model ability to capture relationships among various types of entities. Furthermore, to explore low-order associations of biological bind relations, we introduce a bind-relation enhancement module which can reconstruct missing feature of bind-relation entities and thus generate harder negative sample than random selection. Experimental results conducted on different biological datasets show that the proposed method achieves superior performance compared to advanced baselines, demonstrating its effectiveness and robustness in handling various biological entity relationships. More importantly, the obtained association patterns for the relationship of drug, microbe, and disease are quantitatively visualized with the following biological verification in detail.

Our main contributions are summarized as follow:

- We propose a novel association pattern-aware fusion method for biological entity relationship prediction. The introduce of association pattern-aware strategy can enhance the representation of complex interactions by aggregate features with potential association patterns.

- A bind-relation enhancement module is devised to acquire low-order associations that reveal the biological bind relations, which is essential for reconstructing missing bind-relation entity features and generating challenging negative triplets to enhance the model training.

- Extensive experiments are conducted to verify the superiority of Pattern-BERP, demonstrating its robustness for various biological scenarios. Uniquely compared to the other methods, the interpretability of association patterns is explained to reveal intrinsic biological mechanisms.

## 2 Related Work

In this section, we elaborate on the related work from two distinct yet interconnected perspectives: Biological Entity Relationship Prediction and Network Search and Mining. Each perceptive represents a fundamental aspect of our research, addressing specific challenges and methodologies in applying machine learning techniques to the prediction task.

**Biological Entity Relationship Prediction.** The latest advancements in artificial intelligence have motivated researchers to employ deep learning methodologies for predicting triple-wise biological entity associations. Hypergraph neural network (HGNN)-based methods [17–22] have become the mainstream research direction in this field. Tu et al. [17] proposed a deep hyper-network embedding model to preserve both local and global proximities in the embedding space. Building upon this, Jiang et al. [18] incorporated a dynamic hypergraph construction strategy to capture the hidden and important relations in data structures. Zhang et al. [19] developed a self-attention-based graph neural network applicable to homogeneous and heterogeneous hypergraphs with variable hyperedge sizes. Liu et al. [20] proposed a multi-way relation-enhanced hypergraph representation learning method to predict anti-cancer drug synergy. Liu et al. [21] proposed a multi-view contrastive learning-enhanced hypergraph model for drug-microbe-disease association prediction. In addition, Chen and Li [23] attempted to adopt the tensor decomposition strategy to predict which target a drug binds to when administered to a disease, and further proposed a neural tensor network model [24] that seamlessly combines tensor algebra and deep neural networks to effectively capture the complex nonlinear dependencies among drugs, targets, and diseases.

**Network Search and Mining.** Network search and mining techniques, particularly those utilizing path information including random walks [25–28] and meta-path [29–32], have been widely employed to extract local structural information from networks. These methods have found applications in areas such as content recommendation and community detection [33–37]. Brin and Page [25] introduced a classic ranking algorithm PageRank to determine the importance of web pages based on their link structure. Jeh and Widom [26] adopted a similarity measure based on pairwise random walk, which can capture the structural similarity between nodes, and further extended PageRank with a personalized version [27]. Perozzi et al. [28] proposed Deepwalk that leverages local random walk information to learn vertex latent representations based on deep learning techniques. These above methods are applicable to homogeneous networks and cannot fully utilize the rich semantic information in heterogeneous networks. To address the limitation, Sun et al. [29] introduced a meta path-based similarity framework for heterogeneous information networks, which can capture the subtle semantics of similarity among objects of the same type. Dong et al. [30] proposed a deep learning-based heterogeneous network representation learning method that automatically learns hidden meta-path semantics, generating general node embedding representations. Wang et al. [31] introduced a graph-based fraud detection method that addresses the issue of low homophily by integrating label information to generate distinguishable neighborhood information. Furthermore within the bioinformatics field, Chen et al. [32] proposed a computational algorithm that performs random walks on an integrated network to infer potential relations between proteins and ADRs.

In line with the above methods, Pattern-BERP utilizes path information for triple-wise heterogeneous biological network mining. By leveraging the fusion of association patterns, it facilitates message passing among various biological entities, which will be described comprehensively in Section 4.2.

## 3 Problem Formulation

Given three distinct entity types in biological networks, termed as $\mathcal{A} = \{a_1, a_2, \cdots, a_i, \cdots, a_{|\mathcal{A}|}\}$, $\mathcal{B} = \{b_1, b_2, \cdots, b_j, \cdots, b_{|\mathcal{B}|}\}$, and $\mathcal{C} = \{c_1, c_2, \cdots, c_m, \cdots, c_{|\mathcal{C}|}\}$, their Cartesian product $\mathcal{S} = \mathcal{A} \times \mathcal{B} \times \mathcal{C}$ is a set of all possible triple-wise biological entity associations. For simplicity, the relation $\mathcal{A}$-$\mathcal{B}$-$\mathcal{C}$ is used to represent complex relational semantics for biological entities, such as "drug-microbe-disease", "synergistic drug-drug-cell line" or "drug-target protein-adverse reaction".

For each triplet $(a_i, b_j, c_m) \in \mathcal{S}$, we assign a label $p \in \{0, 1\}$. A label of $p = 1$ indicates that the existence of certain association has been confirmed, while $p = 0$ represents an unknown association which denotes that the association is not yet known and could potentially exist. The objective is to develop a credible model that can predict these potential associations from unknown ones.

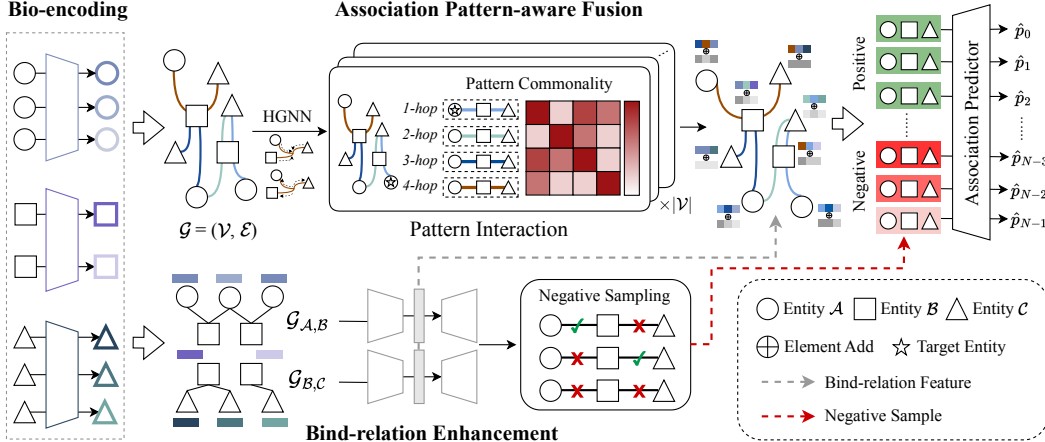

Figure 2: Overall framework of Pattern-BERP. First, entity attributes are initialized with different types of bio-encoders. Then, these existing associations are constructed into one hypergraph $\mathcal{G}$ and two bipartite graphs $\mathcal{G}_{\mathcal{A},\mathcal{B}}$, $\mathcal{G}_{\mathcal{B},\mathcal{C}}$. After that, the hypergraph is encoded with Association Pattern-aware Fusion module based on the pattern commonality, thereby affecting target entity representation. In addition, the bipartite graphs are encoded to output the missing bind-relation feature and thus generate hard negative samples. Finally, the integrated entity feature are used for final association prediction.

## 4 Pattern-BERP

To enhance the ability to represent complex interactions, we propose the first association pattern-aware method, termed as *Pattern-BERP* to extract the rich semantic information embedded within the intricate structures of biological networks. As illustrated in Figure 2, the section is divided into four parts: First, these entity relationships are represented in hypergraph and bipartite graph structures for subsequent module input. Next, the association pattern-aware fusion module is proposed to update entity node feature through association pattern-aware interaction. In addition, the bind-relation enhancement module is introduced to reconstruct bind-relation feature and thus generate hard negative samples. Finally, the detailed summary of loss function and complexity analysis is provided.

### 4.1 Graph Construction

Given the adjacency relationships among entity nodes, the hypergraph and bipartite graph structures are constructed to facilitate the subsequent extraction of structural information and relational association patterns within the respective graphs, which serve for association pattern-aware fusion module and bind-relation enhancement module.

The attributes of each entity are initialized as node features on the graphs with the domain knowledge of bio-entities through its own type. Finally, the initialized node attributes of $\mathbf{X} \in \mathbb{R}^{|\mathcal{V}| \times d}$ consist of features $\mathbf{X}_{\mathcal{A}}$, $\mathbf{X}_{\mathcal{B}}$ and $\mathbf{X}_{\mathcal{C}}$, where $d$ denotes the feature dimensional of initialized entity attributes. Details about entity attribute are provided at Appendix A.1.

**Hypergraph Construction.** Triple-wise biological entity associations can be modeled as a hypergraph $\mathcal{G} = (\mathcal{V}, \mathcal{E})$, which includes a vertex set $\mathcal{V} = \mathcal{A} \cup \mathcal{B} \cup \mathcal{C}$ and a hyperedge set $\mathcal{E} \subset \mathcal{S}$ that represents all known associations. Technically, $\mathcal{G}$ is further formulated as an attributed hypergraph with node attributes $\mathbf{X} \in \mathbb{R}^{|\mathcal{V}| \times d}$ and an incidence matrix $\mathbf{H} \in \{0, 1\}^{|\mathcal{V}| \times |\mathcal{E}|}$, which is defined as:

$$h(v, e) = \begin{cases} 1, & \text{if } v \in e \\ 0, & \text{if } v \notin e. \end{cases} \tag{1}$$

**Bipartite Graph Construction.** The premise of constructing the bind-relation module is to decompose the original triple-wise associations to separately obtain pairwise relations of different types of entities. Since the original triplet is in the form of $(a_i, b_j, c_m)$, and generally the interaction relationships between the three entities are hierarchical (e.g. drug $a_i$ acts on protein $b_j$, and the activated $b_j$ then leads to certain adverse reaction $c_m$). Hence in this paper, we construct two bipartite

graphs: $\mathcal{G}_{\mathcal{A},\mathcal{B}_1}$ for entity $\mathcal{A} \to$ entity $\mathcal{B}_1$[2], $\mathcal{G}_{\mathcal{B}_2,\mathcal{C}}$ for entity $\mathcal{B}_2 \to$ entity $\mathcal{C}$. Then the edge sets of the bipartite graphs can be formulated as:

$$\mathcal{E}_{\mathcal{A},\mathcal{B}_1} = \{(a,b) \mid a \in \mathcal{A}, b \in \mathcal{B}_1, \exists\, e \in \mathbf{H} \text{ such that } a, b \in e\}, \tag{2}$$

$$\mathcal{E}_{\mathcal{B}_2,\mathcal{C}} = \{(b,c) \mid b \in \mathcal{B}_2, c \in \mathcal{C}, \exists\, e \in \mathbf{H} \text{ such that } b, c \in e\}. \tag{3}$$

Note that, the relationship of $\mathcal{A} \to \mathcal{C}$ is not under consideration due to there is no direct connection in this context. The constructed bipartite graphs $\mathcal{G}_{\mathcal{A},\mathcal{B}_1}, \mathcal{G}_{\mathcal{B}_2,\mathcal{C}}$ serve as input for bind-relation module.

## 4.2 Association Pattern-aware Fusion

To comprehensively account for the feature interactions within association patterns, the proposed Association Pattern-aware Fusion (APF) method comprises two fundamental components: Firstly, Association Pattern Sampling (APS) block is designed to sample association patterns by utilizing the distance tokens relevant to target nodes. Secondly, Association Pattern-aware Interaction (API) block is introduced to update node features by message interaction within the sampled association patterns and mine the intrinsic pattern commonality with biological support.

### 4.2.1 Association Pattern Sampling

**Definition 1.** Given certain entity node, the distance token between the node and one hyperedge or association pattern is defined as *u-hop*. Here, *1-hop* patterns represent the hyperedges directly covering the node, *2-hop* patterns represent the hyperedges directly covering all the *1-hop* neighbor nodes of the node, and the relation continues for higher hop counts. If the node is unreachable when $u$ reaches the max step $U$, we define these patterns as no-relation and set the distance to $-\infty$.

Based on the above definition[3], we generate the distance tokens between each node and all hyperedges, ultimately obtaining a distance matrix $\mathbf{D} \in \mathbb{R}^{|\mathcal{V}| \times |\mathcal{E}|}$, defined as follows:

$$d(v,e) = \begin{cases} u, & \text{if node } v \text{ is } \textit{u-hop} \text{ away from pattern } e \\ -\infty, & \text{if node } v \text{ is unreachable from pattern } e \end{cases} \tag{4}$$

where $d(v,e)$ represents the defined distance from node $v$ to pattern $e$; $u$ indicates the number of hops from node $v$ to pattern $e$.

To represent an entity node with association patterns, the general principle is to prioritize and retain patterns that are closer for each node based on the distance tokens, with a total of $N$ patterns sampled. Formally, given an entity node $v$, let $\mathbf{D}_v \in \mathbb{R}^N$ and $\mathbf{P}_v \in \mathbb{R}^{N \times 3d}$ represent the distance tokens and feature vectors of the selected sampled patterns, where each sampled pattern consists of three entity nodes and the $d$-dimensional feature embeddings of each node are defined within the initial node embeddings $\mathbf{X}$. Then, the output pattern feature $z_v$ for node $v$ considers the relative position of these related patterns, which is thus formulated as $\mathbf{z}_v = \mathbf{P}_v + \text{POS}(\mathbf{D}_v)$, where $\text{POS}(\cdot)$ denotes the position encoding layer that maps from $\mathbb{R}^N \to \mathbb{R}^{N \times (3d)}$. Finally, the integrated embedding $\mathbf{z}_v \in \mathbb{R}^{N \times 3d}$ with distance information is produced for use in the subsequent API block.

### 4.2.2 Association Pattern-aware Interaction

Before put into the API block, we adopt a hypergraph convolution layer implemented by [40] on the constructed hypergraph $\mathcal{G}$ to facilitate neighbor-based information propagation among different entities. Then the updated node feature $\mathbf{X}^*$ after hypergraph convolutions is used to construct the feature of patterns mentioned in the APS block, thereby yielding $\mathbf{z}_v$ for the information interaction of these sampled patterns. Details of hypergraph convolution are provided at Appendix.

**Pattern Interaction.** The API block is designed to search for and extract commonalities among different sampled patterns related to a specific entity node, and thus consists of a composition of Transformer layers [38]. Each Transformer layer has two modules: a multi-head self-attention mechanism (MHA) and a position-wise feed-forward network (FFN). For simplicity, we consider the single-head setting, and the extension to multi-head attention is standard and straightforward. Specifically, let $\mathbf{Z}^{(0)} = [\mathbf{z}_1^{(0)}, \ldots, \mathbf{z}_{|\mathcal{V}|}^{(0)}] \in \mathbb{R}^{|\mathcal{V}| \times N \times (3d)}$ denotes the input of the self-attention module

---

[2]To facilitate distinction, we set the entity term $\mathcal{B}_1 = \mathcal{B}_2 = \mathcal{B}$.

[3]The definition is analogous to positional embedding in Transformer [38] and ViT [39].

where $\mathbf{z}_v \in \mathbb{R}^{N \times (3d)}$ is the representation for node $v$. The input $\mathbf{z}_v$ is projected by three matrices $\mathbf{W}_Q \in \mathbb{R}^{(3d) \times d_K}$, $\mathbf{W}_K \in \mathbb{R}^{(3d) \times d_K}$, and $\mathbf{W}_V \in \mathbb{R}^{(3d) \times d_V}$ to obtain the corresponding query, key, and value representations $\mathbf{Q}_v$, $\mathbf{K}_v$, and $\mathbf{V}_v$ for node $v$:

$$\mathbf{Q}_v = \mathbf{z}_v \mathbf{W}_Q, \mathbf{K}_v = \mathbf{z}_v \mathbf{W}_K, \mathbf{V}_v = \mathbf{z}_v \mathbf{W}_V, \mathbf{A}_v = \frac{\mathbf{Q}_v \mathbf{K}_v^\top}{\sqrt{d_K}}, \mathrm{Attn}(\mathbf{z}_v) = \mathrm{softmax}(\mathbf{A}_v) \mathbf{V}_v, \quad (5)$$

where $\mathbf{A}_v \in \mathbb{R}^{N \times N}$ is a matrix capturing the similarity between queries and keys; $d_K$, $d_V$ denotes the feature dimensional of the key representations $\mathbf{K}_v$ and the value representations $\mathbf{V}_v$[4]. Then we will get the output of the self-attention module $\mathbf{z}'_v \in \mathbb{R}^{N \times (3d)}$. To summarize the process of transformer layer, the output of association pattern-aware attention block is computed as:

$$\mathbf{Z}'(l) = \mathrm{MHA}(\mathrm{LN}(\mathbf{Z}^{(l-1)})) + \mathbf{Z}^{(l-1)}, \mathbf{Z}^{(l)} = \mathrm{FFN}(\mathrm{LN}(\mathbf{Z}'(l))) + \mathbf{Z}'(l), \quad (6)$$

where LN denotes layer normalization, and $\mathbf{Z}^{(l)}$ is the output of the current transformer layer. After $L_1$-layer transformers, we get the encoding output $\mathbf{Z}^{(L_1)} = [\mathbf{z}_1^{(L_1)}, \ldots, \mathbf{z}_{|\mathcal{V}|}^{(L_1)}] \in \mathbb{R}^{|\mathcal{V}| \times N \times (3d)}$, and then apply the mean function to similar entities in $N$ association patterns for each node $v$ to obtain the learned embedding $\mathbf{z}_v \in \mathbb{R}^d$.

**Pattern Commonality.** To mine the commonality of $N$ sampled patterns, a score is defined to represent the quantitative relation of these association patterns, termed by *Pattern Commonality Coefficient*, based on the attention scores $\mathbf{A}_v$ of trained API block, which is formulated as follows:

$$\tilde{\mathbf{A}}_v = \mathrm{softmax}(\mathbf{A}_v), \mathbf{C}_v = \frac{1}{N} \sum_{n=1}^{N} \tilde{\mathbf{A}}_v[n, :], \quad (7)$$

where $\mathbf{C}_v \in \mathbb{R}^N$ and $\mathbf{C}_v[n] \in (0, 1)$ indicates the commonality coefficient of the $n$-th pattern. Patterns with relatively high commonality coefficients tend to share the same or similar pathways, while showing significant differences in response compared to low commonality patterns. Corresponding biological validations are presented in the Section 5.3.

### 4.3 Bind-relation Enhancement

In the context of multi-entity relationships, there often exist strong pairwise bind relation between entities, such as drug Aspirin to treat common cold [41, 42]. To mitigate the weakening or overlooking of low-order bind relation, a Bind-relation Enhancement (BE) module is designed to effectively reconstruct the missing feature by capturing these important pairwise associations. Additionally, the module can generate confident and challenging negative samples to aid the training.

**Bind-relation Feature Reconstruction.** To efficiently learn entity representations in pairwise bind relations, we introduce an edge prediction classification task on the bipartite graphs $\mathcal{E}_{\mathcal{A},\mathcal{B}}, \mathcal{E}_{\mathcal{B},\mathcal{C}}$ with the initial embeddings $\mathbf{X}_{\mathcal{A}}^{(0)} = \mathbf{X}_{\mathcal{A}}, \mathbf{X}_{\mathcal{B}_1}^{(0)} = \mathbf{X}_{\mathcal{B}_2}^{(0)} = \mathbf{X}_{\mathcal{B}}, \mathbf{X}_{\mathcal{C}}^{(0)} = \mathbf{X}_{\mathcal{C}}$. Specifically, a $L_2$-layer self-supervised BGNN model [43] is employed to learn node features on the bipartite graphs, followed by the Multi-layer Perception (MLP) [10] layer to output association probabilities, which is defined as:

$$\mathbf{X}_{\mathcal{A}}^{(l+1)}, \mathbf{X}_{\mathcal{B}_1}^{(l+1)} = \mathrm{BGNN}(\mathbf{X}_{\mathcal{A}}^{(l)}, \mathbf{X}_{\mathcal{B}_1}^{(l)}, \mathcal{G}_{\mathcal{A},\mathcal{B}_1}), \mathbf{X}_{\mathcal{B}_2}^{(l+1)}, \mathbf{X}_{\mathcal{C}}^{(l+1)} = \mathrm{BGNN}(\mathbf{X}_{\mathcal{B}_2}^{(l)}, \mathbf{X}_{\mathcal{C}}^{(l)}, \mathcal{G}_{\mathcal{B}_2,\mathcal{C}}), \quad (8)$$

$$\hat{p}_{(a,b_1)} = \mathrm{MLP}(\mathbf{x}_a^{(L_2)} \| \mathbf{x}_{b_1}^{(L_2)}), \hat{p}_{(b_2,c)} = \mathrm{MLP}(\mathbf{x}_{b_2}^{(L_2)} \| \mathbf{x}_c^{(L_2)}), \quad (9)$$

where $\mathbf{X}_{\mathcal{A}}^{(l)}, \mathbf{X}_{\mathcal{B}_1}^{(l)}, \mathbf{X}_{\mathcal{B}_2}^{(l)}, \mathbf{X}_{\mathcal{C}}^{(l)}$ represent the node features at layer $l$; $\mathbf{x}_a^{(L_2)}, \mathbf{x}_{b_1}^{(L_2)}, \mathbf{x}_{b_2}^{(L_2)}, \mathbf{x}_c^{(L_2)}$ denote the final learned representations of entity $a, b_1, b_2, c$ respectively; $\hat{p}_{(a,b_1)}, \hat{p}_{(b_2,c)}$ represent the estimated probability of association between entities $a, b_1$ and entities $b_2, c$ respectively. After that, the loss of the supervised prediction task can be formulated as:

$$\mathcal{L}_{\mathcal{A},\mathcal{B}_1} / \mathcal{L}_{\mathcal{B}_2,\mathcal{C}} = -\frac{1}{|\mathcal{E}_{\mathcal{A},\mathcal{B}_1}| / |\mathcal{E}_{\mathcal{B}_2,\mathcal{C}}|} \sum_{e \in \mathcal{E}_{\mathcal{A},\mathcal{B}_1} / \mathcal{E}_{\mathcal{B}_2,\mathcal{C}}} (p_e \log \hat{p}_e + (1 - p_e) \log (1 - \hat{p}_e)). \quad (10)$$

The loss $\mathcal{L}_{BE}$ of bind-relation task is defined as $\mathcal{L}_{BE} = \alpha \mathcal{L}_{\mathcal{A},\mathcal{B}_1} + (1 - \alpha) \mathcal{L}_{\mathcal{B}_2,\mathcal{C}}$, where $\alpha$ is the balancing coefficient for the two losses.

---

[4]In the single-head setting, $d_K = d_V = 3d$.

**Hard Negative Sampling.** Based on the above bind-relation task, negative samples are adaptively generated for triple-wise associations, instead of randomly selecting from the vast sample space. Moreover, the generated negative samples are challenging, which contributes to efficient learning. As illustrated in Figure 2, three kinds of negative samples are considered as follows:

$$\mathcal{E}_\otimes = \{(a, b', c) \mid (p_{(a,b')} < \gamma) \vee (p_{(b',c)} < \gamma)\}, \tag{11}$$

where $b' \in \mathcal{B}$ represents another entity with random selection that differs from entity $b$ in the original triplet to form the negative sample $(a, b', c)$; $\mathcal{E}_\otimes$ is the set of generated negative triplets; $\gamma$ represents the threshold for the prediction probability to determine whether the association exists.

### 4.4 Total Loss

Base on the above modules, $\mathbf{z}_v$ of node $v$ learned by the APF module is updated with reconstructed features $\mathbf{x}_v$ from the BE module according to the entity type, thereby generate the enhanced embedding of $v$ with $\mathbf{z}_v^* = \mathbf{z}_v + \mathbf{x}_v$ for the association predictor network. Hence, for the triple-wise association prediction, we utilize the learned embeddings $\mathbf{z}_a^*$, $\mathbf{z}_b^*$, and $\mathbf{z}_c^*$ of entity $a$, $b$, $c$ to output the probability of the association $\hat{p}$ through a scoring function $\hat{p}_{(a,b,c)} = \mathrm{MLP}(\mathbf{z}_a^* \parallel \mathbf{z}_b^* \parallel \mathbf{z}_c^*)$. The loss of the association prediction task for true sample set $\mathcal{E}$ and negative sample set $\mathcal{E}_\otimes$ can be formulated as:

$$\mathcal{L}_{APF} = -\frac{1}{|\mathcal{E} \cup \mathcal{E}_\otimes|} \sum_{e \in (\mathcal{E} \cup \mathcal{E}_\otimes)} (p_e \log \hat{p}_e + (1 - p_e) \log (1 - \hat{p}_e)). \tag{12}$$

Hence, the total loss $\mathcal{L}$ is computed through an alternating training strategy of the two modules, where the BE module is trained for the first 4 epochs of every 5-epoch cycle, followed by the APF module, which is trained for the final epoch of each cycle. During the training of each module, the parameters of the other one are frozen. The equation for $\mathcal{L}$ is defined as follows:

$$\mathcal{L}(o) = \mathcal{L}_{BE}(o) \cdot (1 - \left\lfloor \frac{o \mod 5}{4} \right\rfloor) + \mathcal{L}_{APF}(o) \cdot \left\lfloor \frac{o \mod 5}{4} \right\rfloor, \tag{13}$$

where $o$ represents the current epoch; $\lfloor \cdot \rfloor$ denotes floor function; $\mod$ denotes modulo operation.

### 4.5 Complexity Analysis

Considering the significantly higher complexity of the APF module in comparison to other network components, we only consider APF which aggregates multiple patterns across various entity nodes. Specifically, for one entity node with $N$ sampled patterns from the APS module, the input and hidden features in the MHA layers are of dimension $f_M$, and hidden features in the FFN layers are $f_F$. In APF, the query, key, and value matrices are derived from the same input sequence and share length $N$. The primary operations for APF include scaled dot-product attention, multiplication of attention weights and values, MHA linear transformation, and FFN linear projection. The time complexity is expressed as $\mathcal{O}(N^2 \cdot f_M + N \cdot f_M^2 + N \cdot f_M \cdot f_F)$. Hence, for the entire graph, the total complexity is $\mathcal{O}\left(|\mathcal{V}| \cdot (N^2 \cdot f_M + N \cdot f_M^2 + N \cdot f_M \cdot f_F)\right)$, where $|\mathcal{V}|$ is the number of entity nodes.

## 5 Experiment

### 5.1 Experimental Settings

**Datasets.** In this paper, we adopt three biological entity association datasets with significant biological meaning, namely DMD (Drug-Microbe-Disease), DDC (synergistic Drug-Drug-Cell line) and DPA (Drug-target Protein-Adverse reaction), among which DPA dataset is directly constructed. In line with DMD and DDC, we utilize preprocessing tools provided by [21] to deal with the original data from ADReCS-Target [44], and collect a total of 1,079 triplets that are structured into the data schema <drug, protein, adr>. Appendix Table 3 presents the statistics and characteristics of these datasets.

**Baselines.** To verify the effectiveness of Pattern-BERP, we compare it with three kinds of methods: (1) Non-graph methods. Following the work from [21], five non-graph methods are adopted including Random Forest (**RF**) [45], **MLP** [10], **CP** [11], **Tucker** [11], **CoSTCo** [12]; (2) Graph-based methods. We select four classical architectures of graph neural network, including **GCN** [13], **GraphSAGE** [14], **GAT** [15], **GIN** [16]; (3) Hypergraph-based methods. Recent hypergraph learning methods to address triple-wise biological entities associations or similar tasks are considered as the baselines, including **DHNE** [17], **HyperSAGNN** [19], **HGSynergy** [20], **MCHNN** [21].

Table 1: Performance comparison on three datasets of different biological entity associations. Each result of these methods is from the average of 5-fold cross-validation experiments with four scenarios. The best result for each dataset and metric is marked in **bold**. All the presented `hits` scores are in %.

| Methods | DMD | | | DDC | | | DPA | | |
|---|---|---|---|---|---|---|---|---|---|
| | hits@1 | hits@3 | hits@5 | hits@1 | hits@3 | hits@5 | hits@1 | hits@3 | hits@5 |
| RF | 34.06 | 57.28 | 68.31 | 8.93 | 18.63 | 28.29 | 26.62 | 37.45 | 44.95 |
| MLP | 42.72 | 65.40 | 75.33 | 13.27 | 27.98 | 39.88 | 27.55 | 40.65 | 48.80 |
| CP | 44.73 | 66.74 | 76.47 | 13.71 | 28.67 | 41.03 | 31.30 | 45.46 | 54.03 |
| Tucker | 45.27 | 66.54 | 76.98 | 13.24 | 28.55 | 39.77 | 28.80 | 43.10 | 50.74 |
| CoSTCo | 38.69 | 60.38 | 71.77 | 10.93 | 22.95 | 33.60 | 31.06 | 41.95 | 47.87 |
| GCN | 62.66 | 76.57 | 79.74 | 25.86 | 47.06 | 58.27 | 18.38 | 30.56 | 38.66 |
| GraphSAGE | 56.98 | 73.52 | 77.42 | 22.23 | 41.73 | 52.98 | 12.36 | 23.84 | 31.48 |
| GAT | 47.13 | 67.90 | 75.24 | 21.60 | 43.91 | 54.41 | 21.53 | 33.24 | 41.53 |
| GIN | 40.08 | 60.93 | 69.29 | 12.68 | 28.41 | 37.52 | 16.44 | 30.05 | 39.12 |
| DHNE | 81.86 | 93.66 | 96.02 | 43.42 | 62.61 | 72.01 | 32.64 | 47.69 | 56.30 |
| HyperSAGNN | 87.04 | 93.99 | 96.09 | 41.31 | 66.38 | 76.29 | 33.24 | 49.58 | 58.38 |
| HGSynergy | 88.68 | 92.10 | 94.51 | 41.07 | 60.74 | 70.76 | 28.19 | 40.36 | 48.44 |
| MCHNN | 90.04 | 93.92 | 95.19 | 41.91 | 61.12 | 72.10 | 32.27 | 43.29 | 50.32 |
| Pattern-BERP | **93.94** | **97.53** | **98.24** | **48.01** | **68.40** | **76.39** | **43.52** | **57.36** | **63.89** |
| Δ | +3.90 | +3.54 | +2.15 | +4.59 | +2.02 | +0.10 | +10.28 | +7.78 | +5.51 |

**Implementation Details.** To accurately evaluate model performance and prevent overfitting, 5-fold cross-validation is used to evaluate the performance. Specifically, we randomly split the dataset into a 90% cross-validation set and a 10% independent test set. On the cross-validation set, the 5-fold cross-validation is implemented. Moreover, the independent testing, in which the model is trained on the cross-validation set and tested on the independent test set, is conducted to obtain the prediction results. In the training stage, Binary Cross Entropy loss is adopted to measure model performance and Adam optimizer is adopted to optimize all of model parameters with a learning rate of 0.001[5].

**Evaluations.** To evaluate the prediction performance on the triplet associations, hit ratio (hit@n) and normalized discounted cumulative gain (ndcg@n), which are widely used in recommendation tasks [46, 47], are employed for model ability to provide a comprehensive assessment for the task.

## 5.2 Performance Comparison with Advanced Baselines

Table 1 summarizes the prediction performances of Pattern-BERP in comparison with other baselines across the DMD, DDC and DPA datasets. It is evident that Pattern-BERP significantly surpasses the previous state-of-the-art baselines, including non-graph, graph-based, and hypergraph-based methods, across all three datasets by a large margin, with a particularly notable hit@1 improvement of approximately 23.6% on the DPA dataset (from 33.24 to 43.52). The results underscore the broad accuracy and applicability of the proposed method in various biological association scenarios.

A salient observation is that hypergraph-based methods achieve superior performance than those of the other two types. This phenomenon empirically demonstrates the advantages of utilizing high-order structure information over the other methods that we compare. Additionally, we observe that graph-based methods exceed non-graph methods for DMD and DDC datasets. The relatively large number of associations and proportion of DMD and DDC datasets, as shown in Appendix Table 3, indicate a higher density and stronger interconnectivity within the underlying entity relationships. Hence, graph-based methods acquire abundant information of entity interactions based on the graph structure. Furthermore, hypergraph-based methods acquire high-order structure information, ultimately leading to better performance. In contrast, for the DPA dataset, non-graph methods outperform graph-based methods, even demonstrating competitive performance compared to hypergraph-based methods. This can be attributed to the fact that the association proportion within the DPA dataset is conspicuously low (around 0.002%), indicating a relatively sparse graph structure. Under such conditions, the non-graph methods are able to obtain more robust association information compared to graph-based models, which may struggle to capture meaningful patterns from the limited graph connectivity.

---

[5]More details about the experiments can be found at Appendix B.

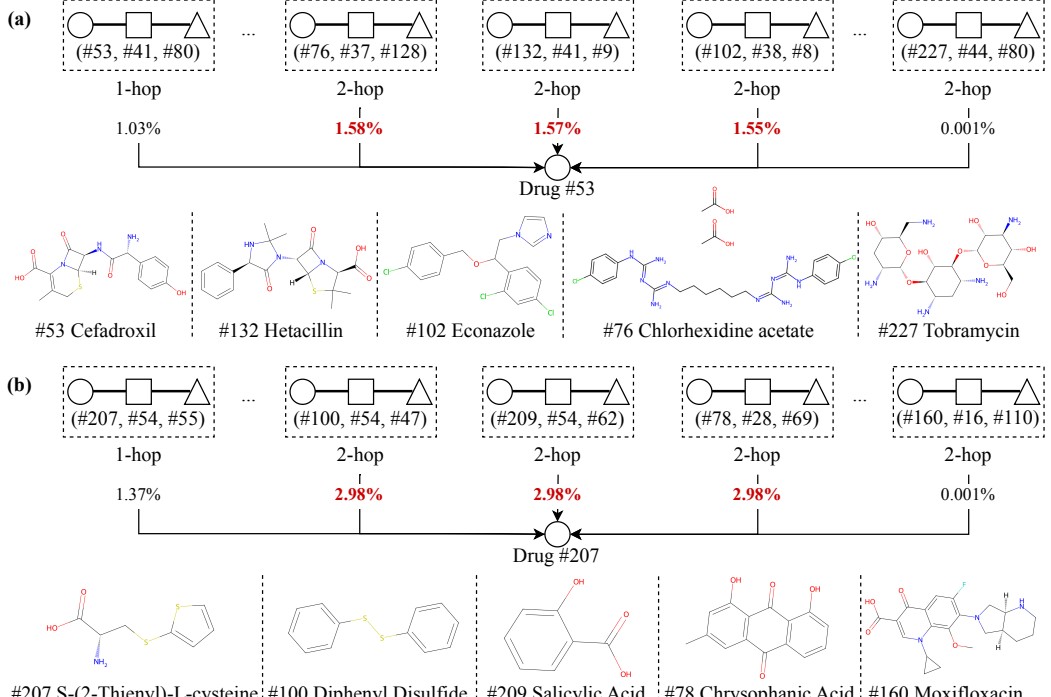

Figure 3: The interpretability cases of $N$=100 association patterns related to drug #53 and #207 in DMD dataset. The pattern commonality coefficients are represented in the form of a percentage to indicate the contribution for visualization, with each pattern typically assigned a default value of 1%. Larger pattern commonality coefficients indicate a more significant contribution to the target drugs, and these patterns frequently exhibit similar or even identical biological pathways. Conversely, smaller commonality coefficients suggest a lack of relevance to these drugs.

## 5.3 Association Pattern Interpretability

In order to investigate the potential relation among different association patterns, we visualize the pattern commonality coefficients of Cefadroxil (drug #53) and S-(2-Thienyl)-L-cysteine (drug #207) in DMD dataset, as shown in Figure 3. Larger pattern commonality coefficients contribute more to the original drug pathway, whereas smaller coefficients often relate to different biological mechanisms. It can be observed that:

- For Cefadroxil (drug #53), the common patterns consisted by Chlorhexidine acetate (drug #76) [48], Hetacillin (drug #132) [49], and Econazole (drug #102) [50, 51] are considered more similar as the mechanism [52] that actives on the cell wall and envelope leading the change of microbe physiological activities, thus treating the diseases. Instead, Tobramycin (drug #227) inhibits mRNA be translated into protein and thus promotes microbe cell death [53, 54].

- For S-(2-Thienyl)-L-cysteine (drug #207), the common patterns consisted by Diphenyl Disulfide (drug #100) [55, 56], Salicylic Acid (drug #209) [57], and Chrysophanic Acid (drug #78) [58, 59] are considered more similar as the mechanism that [60–62] inducts oxidative stress in bacteria and thus damage all components of the microbe cell. Instead, Moxifloxacin (drug #160) inhibits DNA gyrase and topoisomerase IV enzymes to separate bacterial DNA, thereby inhibiting cell replication [63, 64].

In summary, the analyzed cases illustrate the high-commonality patterns derived from Pattern-BERP, demonstrating how these small molecule drugs influence microbial activities to treat diseases, generally following a consistent physiological route. These cases indicate that our method can discover potential common patterns through the API block, and obtain larger pattern coefficients through the attention mechanism, which drives the nodes to learn these important common interactions and acquire more expressive representations. Details about additional cases can be found at Appendix B.3.2.

Table 2: Ablation study results on DDC dataset with different module designs. "BFR" denotes Bind-relation Feature Reconstruction; "HNS" denotes Hard Negative Sampling; "HC" denotes Hypergraph Convolution block; "DE" denotes Distance Embedding; "API" denotes Association Pattern-aware Interaction block. All the presented scores are in %, and the best result is marked in **bold**.

| w/o BFR | w/o HNS | w/o HC | w/o DE | w/o API | hits@1 | hits@3 | hits@5 | ndcg@1 | ndcg@3 | ndcg@5 |
|---|---|---|---|---|---|---|---|---|---|---|
| ✓ | ✓ | - | - | - | 44.76 | 64.74 | 73.08 | 44.76 | 56.42 | 59.85 |
| ✓ | - | - | - | - | 45.03 | 65.86 | 74.97 | 45.03 | 57.19 | 60.95 |
| - | ✓ | - | - | - | 42.81 | 64.48 | 74.02 | 42.81 | 55.41 | 59.33 |
| - | - | - | ✓ | - | 40.29 | 62.87 | 72.44 | 40.29 | 53.41 | 57.35 |
| - | - | - | ✓ | ✓ | 44.69 | 65.17 | 74.51 | 44.69 | 56.62 | 60.45 |
| - | - | ✓ | - | - | 47.11 | 66.56 | 74.89 | 47.11 | 58.47 | 61.90 |
| - | - | - | - | - | **48.01** | **68.40** | **76.39** | **48.01** | **59.84** | **63.13** |

## 5.4 Ablation Study

To investigate the necessity of each component in Pattern-BERP, we conduct several comparisons between Pattern-BERP and its variants on the test set. As illustrated in Table 2, when basic components of Pattern-BERP have been removed, the performances of corresponding variants on DDC dataset significantly decline, indicating that these components all contribute to the performance. Besides, we have other observations: (1) when only DE module is removed, the performance is inferior to the one removing the entire APA module containing DE, demonstrating that inaccurate entity distance information has a more detrimental impact on the prediction performance; (2) eliminating HNS module results in a significant drop in the performance, highlighting its crucial role in enhancing the model's robustness and discrimination capability and thus indicating that the proposed negative sampling strategy contributes to efficient learning; (3) when removing HC module leads to a slight performance degradation, the impact is relatively limited which suggests that the capability of HC module in representing complex associations is relatively modest for the highly dense-association DDC dataset, but it still provides some beneficial effects towards the final performance improvement.

To verify the affect of hyperparameter settings to model performance, we first conduct ablation experiments on the three main parameters of the APF module, namely number of attention heads, number of max pattern distance, and number of sampled association patterns. As illustrated in Appendix Figure 5, when the three hyperparameters are increased, the prediction performance on DDC dataset exhibits an overall upward trend, suggesting that increasing these hyperparameters helps the model better capture the complex association patterns, thereby improving the final performance. Additionally, we conduct experiments on the loss-balanced coefficient $\alpha$ and the bind-relation prediction probability threshold $\gamma$, both adjusted from 0.1 to 0.9. Results in Appendix Figure 6 show that setting $\alpha$ and $\gamma$ to 0.5 yields the best performance. Specifically, for $\alpha$, since the final prediction task involves predicting the associations among entity $\mathcal{A}$, $\mathcal{B}$, and $\mathcal{C}$, the two tasks of $\mathcal{A} \to \mathcal{B}$ and $\mathcal{B} \to \mathcal{C}$ are intuitively of equal importance, therefore the balanced coefficient $\alpha$ set to 0.5; for $\gamma$, bind-relation prediction is fundamentally a binary classification task, thus the threshold $\gamma$ is set to 0.5. Details about ablation experiments are presented at Appendix B.3.3.

## 6 Conclusion

In this work, we propose a novel association pattern-aware fusion method Pattern-BERP for biological entity relationship prediction, which effectively combines the related association pattern information into entity representation learning. In addition, to enhance the missing information of the low-order message passing, we devise a bind-relation module that considers the strong bind of low-order entity associations. The evaluation on three biological datasets quantitatively demonstrate that the proposed method consistently achieve superior performance over the competing baselines. Moreover, the interpretability explanations of association patterns reveal the intrinsic biological mechanisms and thus promote the method to be deployed in real-world scenarios.

Due to the domain-specific task, Pattern-BERP focuses on the fixed-length association patterns. Extending the approach to capture variable-length pathways could further enhance the representational power. Additionally, exploring the applicability of the method in other domains beyond biology, such as general knowledge graph completion, would help evaluate its broader generalizability.

## Acknowledgments and Disclosure of Funding

This work is supported by the Zhejiang Province High-Level Talents Special Support Program "Leading Talent of Technological Innovation of Ten-Thousands Talents Program" (No. 2022R52046), the Fundamental Research Funds for the Central Universities (226-2024-00145), and the Scientific Research Fund of Zhejiang University (No. XY2023020).

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

# Appendix

In this appendix, we provide a comprehensive elaboration of the methodologies, experimental details, and additional insights that support the findings presented in the main manuscript. The appendix is structured into details of the proposed method, details of the experiments, and other discussion contents including limitation and extension. Furthermore, our data, code and all raw experimental results are provided in the Github repository https://github.com/hry98kki/PatternBERP.

## A    Details of Pattern-BERP Method

### A.1    Entity Attribute

Before constructing the graphs, the attributes of each entity are initialized as node features on the graphs with the domain knowledge of bio-entities. Due to the varying types of entities being studied, multiple encoders are employed to align the attribute embeddings for different types of entities. Hence, three specific types of encoders are considered as follows:

- For drug entities $\mathcal{D}$, their SMILES strings can be converted into molecular graphs $\mathbf{G}_{\mathcal{D}} = (\mathbf{S}_{\mathcal{D}}, \mathbf{A}_{\mathcal{D}})$, where $\mathbf{S}_{\mathcal{D}}$ is the attribute matrix of all nodes representing the atoms and $\mathbf{A}_{\mathcal{D}}$ is the adjacency matrix of these nodes. Vallina GIN [16] is adopted to learn atom representations and then these atom representations are summarized into a drug-level feature vector through a global max pooling (GMP), which is formulated as:

$$\mathbf{X}_{\mathcal{D}}^{(l+1)} = \mathrm{MLP}^{(l+1)}((\mathbf{A}_{\mathcal{D}} + (1 + \epsilon)\mathbf{I})\mathbf{X}_{\mathcal{D}}^{(l)}), \tag{14}$$

  where $l$ is the current epoch of GIN, $\mathbf{I}$ is the identity matrix, $\epsilon$ is a fixed scalar, and $\mathbf{X}_{\mathcal{D}}^{(0)} = \mathbf{S}_{\mathcal{D}}$. After applying the GMP over all molecular graphs, the features of all drug entities can be compiled into $\mathbf{X}_{\mathcal{D}} \in \mathbb{R}^{|\mathcal{D}| \times d}$, where $d$ represents the dimension of entity attribute vector.

- For protein entities $\mathcal{P}$, taken the one-dimensional protein sequence $\mathbf{T}$ as the input, we first convert the sequence string to an integer vector as the initialized embedding $\mathbf{S}_{\mathcal{P}} \in \mathbb{R}^{\mathcal{T}}$. Then, considering $\mathbf{X}_{\mathcal{P}}^{(0)} = \mathbf{S}_{\mathcal{P}}$, the 1D CNN [65] model is used to extract the protein representation. The propagation mechanism of each CNN layer works as follows:

$$\mathbf{X}_{\mathcal{P}}^{(l+1)} = \sigma(\mathrm{CNN}(\mathbf{X}_{\mathcal{P}}^{(l)}, d_{in}^{(l)}, d_{out}^{(l)}, ks^{(l)})), \tag{15}$$

  where $\mathbf{X}_{\mathcal{P}}^{(l)}, \mathbf{X}_{\mathcal{P}}^{(l+1)}$ are the hidden feature vectors of the $l^{th}$ and $(l+1)^{th}$ CNN layer, respectively; $d_{in}^{(l)}, d_{out}^{(l)}, ks^{(l)}$ are the number of channels in the input, number of channels produced by the convolution and the convolving kernel size of the $l^{th}$ CNN layer; $\sigma(\cdot)$ represents nonlinear activation function, specially ReLU. After CNN layers, the $d$-dimensional feature vectors of all target proteins are denoted as $\mathbf{X}_{\mathcal{P}} \in \mathbb{R}^{|\mathcal{P}| \times d}$.

- For other entities $\mathcal{O}$ (such as diseases), we compile similarity matrices to acquire the initial embedding $\mathbf{S}_{\mathcal{O}} \in \mathbb{R}^{|\mathcal{O}| \times |\mathcal{O}|}$ based on the methods provided by [32, 66–68] and followed by fully-connected networks to obtain the entity representation, denoted as $\mathbf{X}_{\mathcal{O}} \in \mathbb{R}^{|\mathcal{O}| \times d}$. More specifically, for microbe nodes, cell lines and disease nodes, we compile similarity matrices $\mathbf{S}_{\mathcal{M}} \in \{1, 0\}^{|\mathcal{M}| \times |\mathcal{M}|}$, $\mathbf{S}_{\mathcal{L}} \in \mathbb{R}^{|\mathcal{L}| \times |\mathcal{L}|}$ and $\mathbf{S}_{\mathcal{N}} \in \mathbb{R}^{|\mathcal{N}| \times |\mathcal{N}|}$ based on the methods provided by [66–68], which then are transformed into $\mathbf{X}_{\mathcal{M}} \in \mathbb{R}^{|\mathcal{M}| \times d}$, $\mathbf{X}_{\mathcal{L}} \in \mathbb{R}^{|\mathcal{L}| \times d}$ and $\mathbf{X}_{\mathcal{N}} \in \mathbb{R}^{|\mathcal{N}| \times d}$ by fully-connected networks. For ADRs nodes, we first use the co-occurrence of drugs to evaluate ADRs similarity. For two ADRs $i$ and $j$, the Jaccard score is calculated as follows:

$$\mathrm{Jaccard\_score} = \frac{|D_i \cap D_j|}{|D_i \cup D_j|}, \tag{16}$$

  where $D_{\mathrm{i}}$ and $D_{\mathrm{j}}$ denote drug sets that cause ADR $i$ and ADR $j$, respectively. The ADR similarity matrix $\mathbf{S}_{\mathcal{R}} \in \mathbb{R}^{|\mathcal{R}| \times |\mathcal{R}|}$ is constructed according to the Jaccard score, finally denoted as $\mathbf{X}_{\mathcal{R}} \in \mathbb{R}^{|\mathcal{R}| \times d}$.

Based on the above construction rules, we generate the corresponding initial attributes for entity datasets $\mathcal{A}$, $\mathcal{B}$, and $\mathcal{C}$ through the entity types. Finally, the entity attributes $\mathbf{X}$ consists of features $\mathbf{X}_{\mathcal{A}}$, $\mathbf{X}_{\mathcal{B}}$ and $\mathbf{X}_{\mathcal{C}}$. When encountering specific datasets, the three entity types can be substituted into ($\mathcal{A}$, $\mathcal{B}$, $\mathcal{C}$) respectively. For example, $\mathbf{X}$ consists of $\mathbf{X}_{\mathcal{D}}$, $\mathbf{X}_{\mathcal{P}}$ and $\mathbf{X}_{\mathcal{R}}$ in DPA dataset.

Table 3: Detailed information of three datasets of different biological entity association.

| Datasets | Entity Types | | | #Nodes | | | #Associations | Ratio |
|---|---|---|---|---|---|---|---|---|
| DMD | Drugs | Microbes | Diseases | 270 | 58 | 167 | 2,763 | 0.106% |
| DDC | synergistic Drugs | Drugs | Cell lines | 87 | 87 | 55 | 2,044 | 0.491% |
| DPA | Drugs | Proteins | ADRs | 298 | 552 | 280 | 1,079 | 0.002% |

## A.2 Association Pattern Distance

Given the maximum number of hops $U$ (i.e., the max positional distance for certain entity node), the distance matrix $\mathbf{D} \in \mathbb{R}^{|\mathcal{V}| \times |\mathcal{E}|}$ is constructed based on Definition 1 as follows:

- **1-hop distance:** For each node $v$ and hyperedge $e$, if hyperedge $j$ directly contains node $i$, then $\mathbf{D}_{v,e} = 1$ represents the association of hyperedge $j$ is the 1-hop pattern.
- **2-hop distance:** If hyperedge $e$ does not directly contain node $v$ but shares a node with a hyperedge that does, then $\mathbf{D}_{v,e} = 2$ represents the association of hyperedge $e$ is the 2-hop pattern.
- **u-hop distance:** Recursively calculate further distances up to $u$-hop, then $\mathbf{D}_{v,e} = u$.
- **Unreachable:** If node $v$ cannot reach hyperedge $e$ within $U$-hop, then $\mathbf{D}_{v,e} = -\infty$.

Hence, the distance tokens $\mathbf{D}_v$ for node $v$ are transformed by one position encoding layer, thereby yielding the positional embeddings for the self-attention mechanism of the subsequent API block.

## A.3 Theory Analysis

As an empirical method, the Pattern-BERP method extracts common patterns or rules from large bio-associated networks, similar to the k-means clustering. Specifically, given a triplet-wise dataset $\mathcal{S} = \{\mathbf{s}_1, \mathbf{s}_2, \ldots, \mathbf{s}_t, \ldots, \mathbf{s}_T\}$ and $K$ common patterns, the optimization goal is to minimize the total distance of sampled patterns to their respective pattern centers. First, initialize the pattern centers $\mathcal{M} = \{\mathbf{m}_1, \mathbf{m}_2, \ldots, \mathbf{m}_k, \ldots, \mathbf{m}_K\}$. Next, assign each sample to the nearest pattern center by computing the center index with $\arg\min_k \|\mathbf{s}_i - \mathbf{m}_k\|$ for all the centers in $\mathcal{M}$. Then, update the position of these pattern centers using $\mathbf{m}_k = \frac{1}{|C_k|} \sum_{\mathbf{s}_t \in C_k} \mathbf{s}_t$, where $C_k$ is the set of samples assigned to pattern $k$ calculated as above. Iterate these assignment and update steps until the pattern centers converge. The objective function to minimize is $J = \sum_{k=1}^{K} \sum_{\mathbf{s}_t \in C_k} \|\mathbf{s}_t - \mathbf{m}_k\|^2$, and by minimizing $J$, Pattern-BERP extracts representative biological association patterns effectively.

# B  Details of the Experiments

## B.1  Datasets

In this paper, three biological datasets are used to evaluate the efficacy of the proposed method. Each dataset encompasses three different entities and their associations. Appendix Table 3 provides a detailed presentation of the specific entities within each dataset, including the count of nodes per entity, the number of associations among these nodes, and the corresponding association ratio. All entity associations are structured into triplet scheme, such as <drug, protein, adr> for DPA dataset.

**DPA dataset.** DPA dataset is first constructed and preprocessed with the origin data from [44]. Specifically, only <drug, target, adr> triplets that have complete field information are retained, that is, drug Pubchem CID, target protein identifier UniProt ID, ADR term name. Then, the detail information is obtained through these unique identifiers, respectively. For drugs, the PubChem database [69] is queried with the PubChem CID of each drug, and their Canonical SMILES are recorded. For proteins, the UniProtKB database [70] is queried with the UniProtKB ID of each protein, and their protein sequences are recorded. For ADRs, the incidence matrix about the reaction of specific ADRs to ~15,000 drugs is obtained from publicly available datasets [71]. In this matrix, a value of 1 indicates that a particular drug causes a specific ADR, while a value of 0 denotes no relation between them. Furthermore, following the method described in Appendix A.1, the ADR similarity matrix can be constructed for subsequent feature generation fo ADR entities.

Table 4: Detailed information of memory usage (**GB**) with varying the number of sampled patterns $N$ across three datasets on a single NVIDIA A6000 GPU with Intel(R) Xeon CPU (24 cores).

| Datasets | Number of Nodes | Number of Associations | $N = 5$ | $N = 10$ | $N = 20$ | $N = 50$ | $N = 100$ |
|---|---|---|---|---|---|---|---|
| DMD | 495 | 2,763 | 10.52 | 10.65 | 10.93 | 11.77 | 13.30 |
| DDC | 229 | 2,044 | 4.35 | 4.40 | 4.58 | 4.96 | 5.64 |
| DPA | 1,130 | 1,079 | 9.68 | 10.01 | 10.62 | 12.55 | 16.01 |

## B.2 Details of Implementation

### B.2.1 Implementation of Baselines

The implementation of all baseline methods is conducted using their respective publicly accessible source codes. Optimal or default configurations for each method are employed to ensure robustness. Specifically, for methods such as RF, MLP [10], CP and Tucker [11], meticulous parameter tuning is engaged in to elicit their peak performance levels. For GCN [13], GAT [15], GraphSAGE [14] and GIN [16], the original triple-wise associations are initially decomposed into two pair-wise associations. Subsequently, each pair-wise association is independently modeled using the corresponding graph neural network model, and the prediction probabilities of the two pair-wise associations are multiplied to obtain the final prediction value for the triple-wise association. Additionally, for DHNE [17] and HyperSAGNN [19], biological embeddings are integrated with their original structural embeddings to ensure fairness. For CoSTCo [12], HGSynergy [20] and MCHNN [21], the parameter settings outlined in the original publications are followed. It is important to note that for the sake of equitable comparison, all of these methods utilize negative sampling setting from MCHNN that is consistent across implementations.

### B.2.2 Implementation Settings

The initialized entity embedding size $d$ is fixed to 128. The number of BGNN, APF layers are all fixed to 2. The training epoch is setting to 1,000 for DMD, DPA datasets and 2,000 for DDC dataset. The number of max hop in pattern sampling $U$ is setting to 3. The number of sampled patterns $N$ is setting to 100. The number of attention heads is 32 for DMD dataset, 16 for DDC dataset, 4 for DPA dataset. The loss-balanced coefficient for bind-relation task $\alpha$ is fixed to 0.5. The threshold for bind-relation prediction probability $\gamma$ is fixed to 0.5. In addition, in line with the evaluation strategy of [21], 29 negative samples are randomly generate for each test triplet in four scenarios for the evaluation of all methods, and thus display the average metrics over all test triplets. These scenario settings can comprehensively evaluate the model ability to identify positive and negative samples under these stringent conditions.

Furthermore, all experiments are conducted on a single NVIDIA A6000 Tensor Core GPU (48GB) and Intel(R) Xeon CPU with 24 cores and 500G memory. The whole training time for DMD, DDC, DPA datasets is about 8, 8, 4 hours, respectively. In addition, Appendix Table 4 exhibits the GPU memory usage with varying the number of sampled patterns $N$ across three datasets, and Appendix Figure 7 presents the inference time of Pattern-BERP in comparison with these advanced baselines for each 100 samples with milliseconds.

## B.3 Details of Results

### B.3.1 Additional Performance Comparison on the ndcg

As a supplement to Table 1, Appendix Table 5 incorporates additional metrics ndcg@1, ndcg@3 and ndcg@5. Considering all these metrics, Pattern-BERP significantly outperforms previous SOTA baselines on all three datasets, exhibiting the remarkable advantages of association pattern mining.

Furthermore, each result of these methods is from the average of 5-fold cross-validation experiments with four scenarios. However, significant variation in prediction difficulty across different scenarios makes it relatively unreasonable to provide an error bar for all 20 results. Instead, to demonstrate the statistical validity of Pattern-BERP, we have provided the raw experimental results.

Table 5: Performance comparison on three datasets of different biological entity associations. Each result of these methods is from the average of 5-fold cross-validation experiments with four scenarios. The best result for each dataset and metric is marked in **bold**. All the presented ndcg scores are in %.

| Methods | DMD | | | DDC | | | DPA | | |
|---|---|---|---|---|---|---|---|---|---|
| | ndcg@1 | ndcg@3 | ndcg@5 | ndcg@1 | ndcg@3 | ndcg@5 | ndcg@1 | ndcg@3 | ndcg@5 |
| RF | 34.06 | 47.51 | 52.08 | 8.93 | 14.50 | 18.45 | 26.62 | 32.86 | 35.94 |
| MLP | 42.72 | 55.88 | 59.99 | 13.27 | 21.73 | 26.63 | 27.55 | 35.13 | 38.47 |
| CP | 44.73 | 57.50 | 61.51 | 13.71 | 22.31 | 27.37 | 31.30 | 39.53 | 43.05 |
| Tucker | 45.27 | 57.64 | 61.92 | 13.24 | 22.03 | 26.61 | 28.80 | 37.10 | 40.23 |
| CoSTCo | 38.69 | 51.21 | 55.93 | 10.93 | 17.73 | 22.13 | 31.06 | 37.43 | 39.86 |
| GCN | 62.66 | 71.00 | 72.31 | 25.86 | 38.07 | 42.70 | 18.38 | 25.39 | 28.71 |
| GraphSAGE | 56.98 | 66.84 | 68.45 | 22.23 | 33.43 | 38.05 | 12.36 | 19.01 | 22.17 |
| GAT | 47.13 | 59.35 | 62.38 | 21.60 | 34.53 | 38.85 | 21.53 | 28.26 | 31.65 |
| GIN | 40.08 | 52.27 | 55.71 | 12.68 | 21.74 | 25.50 | 16.44 | 24.24 | 27.95 |
| DHNE | 81.86 | 88.88 | 89.86 | 43.42 | 54.54 | 58.40 | 32.64 | 41.45 | 45.00 |
| HyperSAGNN | 87.04 | 91.18 | 92.05 | 41.31 | 55.88 | 59.97 | 33.24 | 42.73 | 46.36 |
| HGSynergy | 88.68 | 90.82 | 91.28 | 41.07 | 51.51 | 55.19 | 28.19 | 35.21 | 38.10 |
| MCHNN | 90.04 | 92.38 | 92.91 | 41.91 | 53.03 | 57.54 | 32.27 | 38.59 | 41.49 |
| Pattern-BERP | **93.94** | **96.10** | **96.39** | **48.01** | **59.84** | **63.13** | **43.52** | **51.67** | **54.36** |
| Δ | +3.90 | +3.72 | +3.48 | +4.59 | +3.96 | +3.16 | +10.28 | +8.94 | +8.00 |

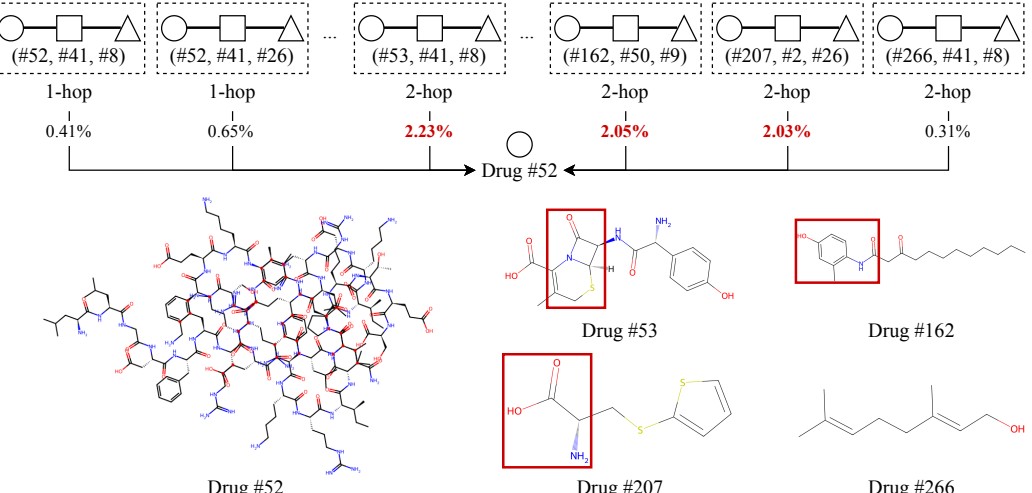

Figure 4: The interpretability case of $N$=100 association patterns related to drug #52 in DMD dataset. The pattern commonality coefficients are represented in the form of a percentage to indicate the contribution for visualization. Larger pattern commonality coefficients indicate a more significant contribution to the target drug #52, and these patterns frequently exhibit similar or even identical biological pathways. Conversely, smaller coefficients suggest a lack of relevance to drug #52.

### B.3.2 Additional Interpretability Analysis

In order to investigate the potential relation among different association patterns, we visualize the pattern coefficients of drug #52 in DMD dataset, as shown in Appendix Figure 4. It can be observed that the patterns that make important contributions are not necessarily 1-hop patterns, with pattern coefficients of 0.41% and 0.65%. In contrast, the 2-hop patterns can exhibit considerable relevance, due to the similar mechanisms of toxicity against microbes exhibited by the corresponding drugs (#53, #162, #207) compared to the target drug #52 under investigation. Additionally, the weakening of drug #52 itself is because the original drug #52 is a complex peptide structure with repetitive and redundant information, thus acquiring simpler and more straightforward representations through the aforementioned information interaction.

Furthermore, from the view of structural and functional groups mentioned by [72–75], we have several salient observations as follows: (1) As the drug in the most influenced pattern, drug #53

Table 6: Ablation study results on DMD dataset with different module designs. "BFR" denotes Bind-relation Feature Reconstruction; "HNS" denotes Hard Negative Sampling; "HC" denotes Hypergraph Convolution block; "DE" denotes Distance Embedding; "API" denotes Association Pattern-aware Interaction block. All the presented scores are in %.

| w/o BFR | w/o HNS | w/o HC | w/o DE | w/o API | hits@1 | hits@3 | hits@5 | ndcg@1 | ndcg@3 | ndcg@5 |
|---|---|---|---|---|---|---|---|---|---|---|
| ✓ | ✓ | - | - | - | 77.67 | 91.72 | 95.16 | 77.67 | 86.06 | 87.48 |
| ✓ | - | - | - | - | 78.96 | 92.71 | 95.98 | 78.96 | 87.15 | 88.52 |
| - | ✓ | - | - | - | 92.85 | 97.10 | 98.06 | 92.85 | 95.39 | 95.80 |
| - | - | - | ✓ | - | 92.83 | 96.94 | 97.88 | 92.83 | 95.31 | 95.70 |
| - | - | - | ✓ | ✓ | 93.76 | 97.38 | 98.09 | 93.76 | 95.95 | 96.24 |
| - | - | ✓ | - | - | 93.59 | 97.27 | 98.06 | 93.59 | 95.97 | 96.18 |
| - | - | - | - | - | 93.94 | 97.53 | 98.24 | 93.94 | 96.10 | 96.39 |

Table 7: Ablation study results on DPA dataset with different module designs. "BFR" denotes Bind-relation Feature Reconstruction; "HNS" denotes Hard Negative Sampling; "HC" denotes Hypergraph Convolution block; "DE" denotes Distance Embedding; "API" denotes Association Pattern-aware Interaction block. All the presented scores are in %.

| w/o BFR | w/o HNS | w/o HC | w/o DE | w/o API | hits@1 | hits@3 | hits@5 | ndcg@1 | ndcg@3 | ndcg@5 |
|---|---|---|---|---|---|---|---|---|---|---|
| ✓ | ✓ | - | - | - | 37.73 | 53.94 | 62.41 | 37.73 | 47.15 | 50.64 |
| ✓ | - | - | - | - | 34.21 | 52.18 | 60.79 | 34.21 | 44.62 | 48.17 |
| - | ✓ | - | - | - | 42.27 | 57.45 | 62.96 | 42.27 | 51.12 | 53.39 |
| - | - | - | ✓ | - | 41.57 | 55.65 | 63.01 | 41.57 | 49.74 | 52.77 |
| - | - | - | ✓ | ✓ | 46.16 | 60.60 | 67.22 | 46.16 | 54.63 | 57.36 |
| - | - | ✓ | - | - | 44.68 | 58.52 | 64.91 | 44.68 | 52.79 | 55.44 |
| - | - | - | - | - | 43.52 | 57.36 | 63.89 | 43.52 | 51.67 | 54.36 |

contains $\beta$-lactam as shown in the red rectangle of Figure 3. $\beta$-lactam is the crucial component of $\beta$-lactam antibiotics [76], which is one of the most widely used classes of antibiotics available. In addition, Islam et al. [77] also prove that pyridine scaffold (aromatic ring with nitrogen) bearing poor basicity generally improves water solubility in pharmaceutically potential molecules and has led to the discovery of numerous broad-spectrum therapeutic agents. (2) Similar with drug #53, drug #162 has acetanilide structure, which serves as the basis for antimicrobial activity and disease treatment [78]. (3) Drug #207 has the dimethylglycine structure, which is crucial to Tigecycline [79]. Tigecycline binds to the bacterial ribosome, blocking the binding of amino-acyl-tRNA to the acceptor site on the mRNA-ribosome complex, thereby inhibiting protein synthesis. (4) Unlike the previous three, drug #266 may interact with membrane lipids to alter membrane fluidity and permeability, thereby exerting its effect on microorganisms. However, this mechanism of action is distinct from the specific functional group interactions of the previous three drugs.

As a compound with a complex structure containing multiple functional groups, drug #52 is similar to the previous three drugs due to its complex structure and diverse functional groups. These characteristics enable it to exert toxicity through specific interactions with the critical pocket of microbes. Compared to drug #266, these complex drugs have more diverse and specific toxicity mechanisms, which is why they have received less attention.

### B.3.3 Ablation Results

**Network Module Designs.** To investigate the necessity of each component in Pattern-BERP, we conduct several comparisons between Pattern-BERP and its variants on the independent test set. Table 2, 6, 7 denotes the results of DDC, DMD, DPA dataset, respectively.

When basic components of Pattern-BERP have been removed, the performances of corresponding variants on all datasets exhibit significant declines, though the degree of performance degradation varies. This indicates that these components all contribute to the performance. Besides, we have other observations: (1) for the relatively sparse datasets, DMD and DPA, the performance decline after removing BFR is more significant compared to the DDC dataset. (2) for all datasets, eliminating HNS module results in a drop in the performance, highlighting its effect in enhancing the model's robustness

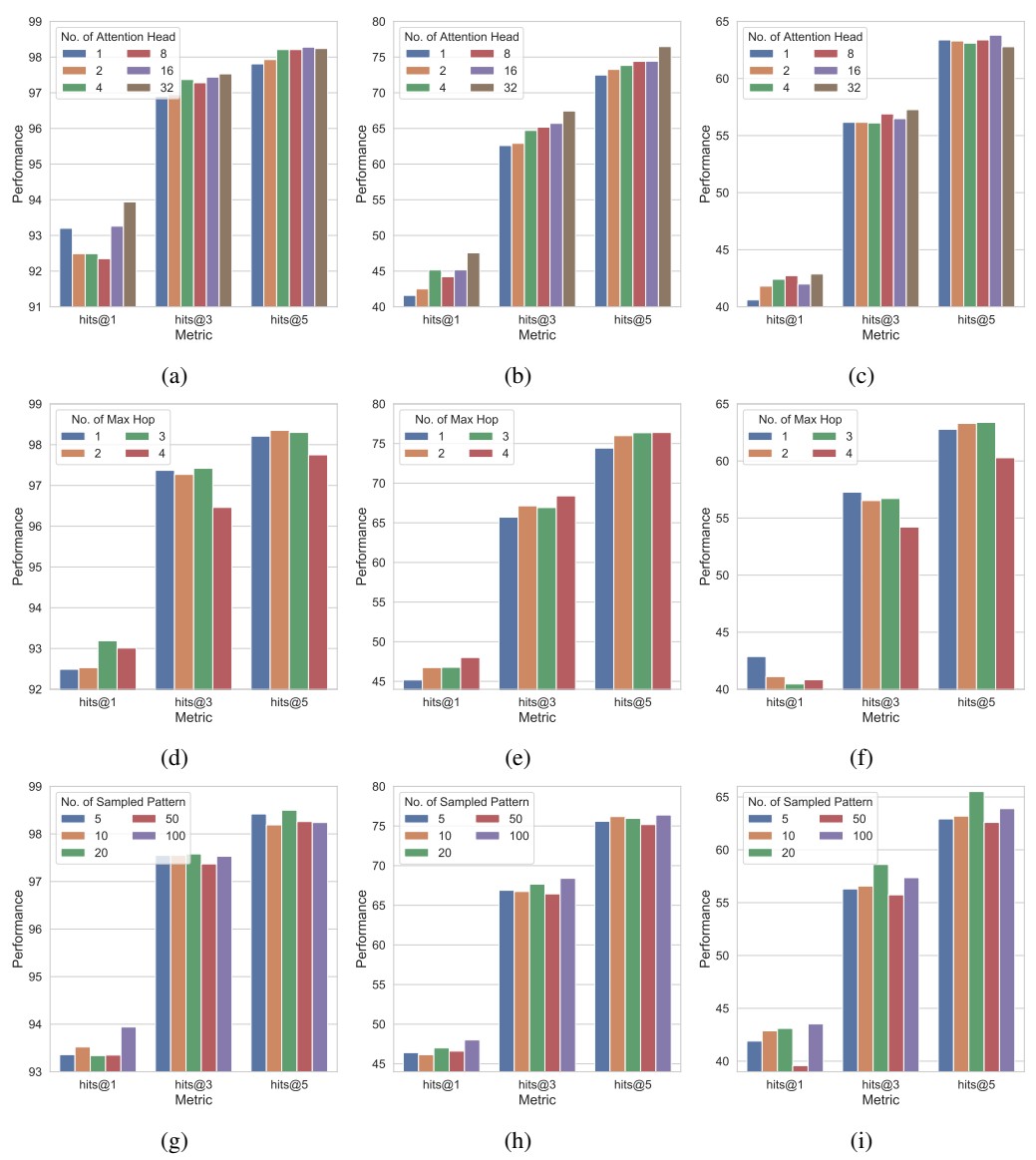

Figure 5: Ablation study results with different hyperparameter settings. The three rows are for DMD, DDC, DPA dataset, respectively.

and discrimination capability and thus indicating that the proposed sampling strategy contributes to efficient learning; (3) for all datasets, when only DE module is removed, the performance is inferior to the one removing the entire API module containing DE, demonstrating that inaccurate entity distance information has a more detrimental impact on the prediction performance; (4) when removing HC module leads to a slight performance degradation, the impact is relatively limited which suggests that the capability of HC module in representing complex associations is relatively modest for the relatively dense-association DDC and DMD dataset, but it still provides some beneficial effects towards the final performance improvement. But for highly sparse DPA datasets, removing HC module leads to performance improvement, demonstrating that the combined performance of HC and APA is instead constrained by the confusion introduced by the two aggregation modes on the extremely sparse structures.

**Hyperparamter Settings.** To verify the ablation of hyperparameter settings, we conduct ablation experiments on the three main parameters, namely number of attention heads, number of max pattern distance, and number of sampled association patterns. As illustrated in Appendix Figure 5, when the number of attention head are increased, the prediction performance on all datasets exhibits an

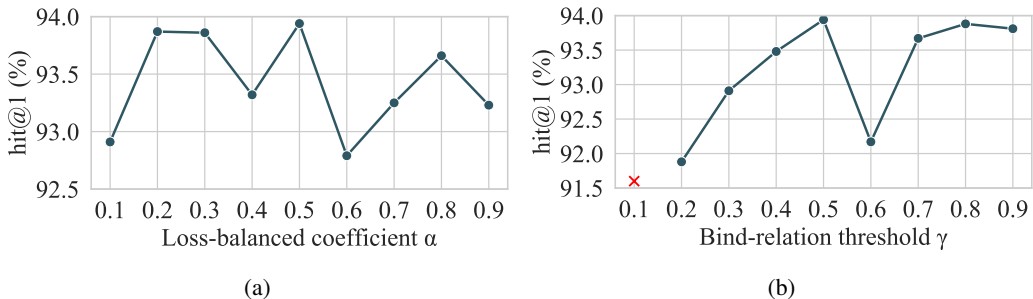

Figure 6: Ablation study results on the loss-balanced coefficient $\alpha$ and the probability threshold of bind-relation task $\gamma$, both varying from 0.1 to 0.9. Note that the red cross in (b) indicates that valid negative samples cannot be generated when $\gamma = 0.1$, hence the predictions cannot be made.

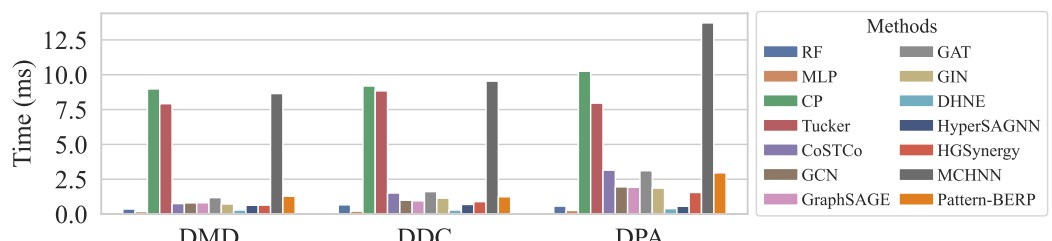

Figure 7: Inference time comparison on three datasets of different biological entity associations. Each result of these methods is for 100 triplet samples. All the presented scores are in milliseconds (**ms**). Note that the original implementation of CP, Tucker and MCHNN is CPU-only based.

overall upward trend, suggesting that increasing the attention heads helps the model better capture the complex association patterns, thereby improving the final performance. In addition, given the significant disparity in the association ratios across the three datasets, the performance of DDC dataset improves as the max hop hyperparameter is increased. In contrast, for the relatively sparse DMD dataset, it is advisable to restrict the max hop to the range from 1 to 3, as the performance drops sharply when the parameter is set to 4. Moreover, for the sparsest DPA dataset, selecting only 1-hop patterns appears to be the optimal choice. Moreover, regarding the number of sampled patterns, a similar trend is observed across the datasets. For DMD and DPA datasets, which are relatively sparse, increasing the number of sampled patterns is not necessarily beneficial for performance. In contrast, on DDC dataset, augmenting the number of sampled patterns continues to yield performance.

Furthermore, we also conduct ablation experiments on another two parameters, namely loss-balanced coefficient $\alpha$ and the bind-relation prediction probability threshold $\gamma$. Results in Appendix Figure 6 show that when $\alpha$ and $\gamma$ are both set to 0.5 by our default, the performance surpasses those of other settings. Specifically, for $\alpha$, since the final prediction task involves predicting the associations among entity $\mathcal{A}$, $\mathcal{B}$, and $\mathcal{C}$, the two tasks of $\mathcal{A} \rightarrow \mathcal{B}$ and $\mathcal{B} \rightarrow \mathcal{C}$ should be intuitively considered to be of equal importance, therefore the balanced coefficient $\alpha$ set to 0.5; for $\gamma$, bind-relation prediction is fundamentally a binary classification task, thus the threshold $\gamma$ is set to 0.5.

## C  Discussions

**Computation Complexity Discussion.** To facilitate applying across a wider range of dataset types and particularly large datasets, it is imperative to rigorously address the issue of complexity explosion. Regarding computation complexity, the discussion in Section 4.5 indicates that the decisive factors are the number of entity nodes $|\mathcal{V}|$ and the number of sampled patterns $N$. Hence, when encountering larger datasets or more complex biological networks, we can employ the following two strategies to avoid a computation complexity explosion: (1) Graph Sampling. By sampling smaller subgraphs [80], computational resource consumption reduces significantly, boosting algorithm speed and efficiency; (2) High-confidence Pattern Selection. As shown in Appendix Figure 5 (g)-(i), reducing $N$ from 100

to 5 across three datasets slightly decreases performance but still surpasses these baselines. Thus, adjusting the sampling quantity is acceptable to effectively reduce time complexity.

Moreover, Appendix Table 4 shows the GPU usage varying from the number of sampled pattern across three datasets with significant differences in the number of entities and associations. The inference time of each 100 samples with milliseconds compared to baselines are provided in Appendix Figure 7. It is evident that the current computational resources are sufficient to handle existing datasets, and the difference in inference time for our method is constant and even faster than several baselines.

**Potential Application in Real-world Scenarios.** The preliminary evaluation results on three biological datasets quantitatively demonstrate that the proposed method consistently achieves superior performance over the competing baselines. Additionally, the learned association patterns show potential in interpreting biological mechanisms. This finding provides hope for the future practical application of our approach, particularly in addressing the "black box" issue in the field of bioinformatics. As a computer-aided tool, our method holds the potential to exhibit a broad array of applications in real-world scenarios, contingent upon sufficient clinical tests or validations. Consequently, it may significantly contribute to human health and well-being.

**Limitations and Future Work.** While the proposed Pattern-BERP has demonstrated strong performance on biological entity relationship prediction, there are several limitations and potential directions for future research. One limitation is that the current model only considers fixed-length association patterns. In real-world biological systems, relevant relationships may be expressed through variable-length pathways involving multiple intermediate entities and relations. Extending the model to capture and leverage such variable-length association patterns could further enhance its representational power and predictive accuracy. Additionally, the applicability of Pattern-BERP method has so far been explored only in the biological domain. Investigating the effectiveness of this approach in other domains, such as knowledge graph completion for general entities or relation extraction from text, would help evaluate its broader generalizability and potential to benefit other fields that rely on structured knowledge representations. Adapting the model architecture and training strategies to accommodate the unique characteristics of different domains could lead to fruitful avenues for future research.

By addressing these limitations and exploring these future research directions, we believe Pattern-BERP can be further improved and extended to have an even greater impact on advancing our understanding of complex biological systems and knowledge representation in general.

