# OpenReview forum: "Association Pattern-aware Fusion for Biological Entity Relationship Prediction"
_NeurIPS.cc/2024/Conference — NeurIPS 2024 poster_

### Official Review · Reviewer_UqdA · 2024-07-04

**Soundness:** 3
**Presentation:** 3
**Contribution:** 3
**Rating:** 7
**Confidence:** 3

**Summary:**

The paper introduces an innovative perspective, namely association pattern, for biological entity relationship prediction (e.g., drug-target protein-adverse reaction). The authors first summarize the characteristics of existing perspectives and emphasize the importance of the association pattern. Next, the association patterns of certain entity nodes are sampled in the constructed graph based on the defined distance and then serve as tokens for self-attention to fuse similar semantic information. Additionally, the bind-relation enhancement module can effectively reconstruct the missing feature and thus generate hard negative samples for further improving the performance. Extensive experiments on biological datasets demonstrate the effectiveness of the proposed method.

**Strengths:**

1. The perspective of triple-wise association patterns, which takes the actual meaning of each entity and their connections into account, is novel and explainable.
2. The paper comprehensively discusses the advantages of the proposed method compared with existing non-graph, graph, and hypergraph perspectives. Moreover, the related works on biological entity relationship prediction and path-based graph mining are elaborated.
3. The paper provides a clear and detailed description of each part of the method, and the formulas used in the paper seem correct. The overall design of the method is rather reasonable.
4. Extensive experiments on biological datasets demonstrate the effectiveness of the proposed method. The reveal of the intrinsic biological mechanism can indeed promote the deployment of the method in real-world scenarios.
5. The association pattern-aware perspective appears to apply to commonly used graph benchmarks or tasks, which can be leveraged to enhance existing graph-based methods.

**Weaknesses:**

1. The bind-relation feature reconstruction appears to be a standalone component that requires extra separate training, resulting in a two-stage process for the method. Additionally, such a design may influence the entire optimization process and distract the training flow, how to alleviate that?

2. Generalization and extension issues have not been discussed. The pattern's length is fixed to 3, which may make the method not work when facing longer-range relationships. Are there any feasible solutions for sampling patterns with different lengths?

3. Complexity has not been discussed. Some details are also not clear enough, such as in Line 167, could the authors provide details on how the position encoding is designed? In Fig. 2, the meaning of the dotted line "Bind-relation Feature" needs more explanation. The "add" operation is not reflected in Section 4.3. Please explain the meaning of the dotted line "Bind-relation Feature".

**Questions:**

Q1: The paper does not give a specific explanation of $L_{BE}$ (see Line 224). Please give a specific explanation of $L_{BE}$.

Q2: In Table 1, the authors appear to have not reported the statistical measures like standard error.

**Limitations:**

The authors have adequately discussed the limitations.

---

> ### Author Rebuttal · Authors · 2024-08-07
>
> We thank the reviewer for all the comments. We believe we have addressed all the concerns and are happy to follow up in the discussion phase.
>
> > W1: The bind-relation feature enhancement module needs extra separate training?
>
> A: Thanks for pointing this out. Our description may not be entirely clear, and the overall architecture is actually quite straightforward. The core components are two complementary modules: Association Pattern-aware Fusion (APF, Section 4.2), Bind-relation Enhancement (BE, Section 4.3) with their corresponding losses.
>
> * Training strategy. Actually, the method does not adopt two completely independent training stages, but rather achieves it through an alternating training strategy. Specifically, the BE module is trained for the first 4 epochs of every 5-epoch cycle, followed by the APF module, which is trained for the final epoch of each cycle. During the training of each module, the parameters of the other one are frozen. Moreover, this design satisfies the need of generating hard negative samples for the APF module, hence the BE module requires initial and additional training.
> * Loss funtions. The BE losses (Eq. 7 & Eq. 8) are utilized to supervise the reconstruction of missing pairwise feature and the generatation of hard negative samples. Consequently, the BE module necessitates additional initial training to prevent low-confidence negative samples for the APF module. Following this, the supervised loss $\mathcal{L}_{APF}$ (Eq. 10) is utilized to determine the positive triplet samples and these negative samples generated from the BE module.
>
> Hence, the total loss function is formulated as  $\mathcal{L}(o) = \mathcal{L}\_{\text{BE}}(o) \cdot (1 - \left\lfloor \frac{o \mod 5}{4} \right\rfloor) + \mathcal{L}\_{\text{APF}}(o) \cdot \left\lfloor \frac{o \mod 5}{4} \right\rfloor$, where $o$ represents the current epoch; $\lfloor \cdot \rfloor$ denotes the floor function; $\text{mod} $ denotes the modulo operation. We will make the description of the entire architecture more clear for understanding.
>
> > W2: Generalization issues about sampling patterns with different lengths.
>
> A: Thanks for advice. As shown in Table R2 (response PDF), the proposed method has been extended with variable-length settings and thus applied on the molecular property prediction tasks. See Q3 of reviewer ugjG for details.
>
> > W3.1: Computation complexity.
>
> A: Thanks for concern. The computation complexity depends on the number of entity nodes and sampled patterns, and we discuss the solution for complexity explosion. Please refer to Q1 & L1 of reviewer ugjG for details.
>
> > W3.2: Details about how the position encoding designs.
>
> A: Thanks for your feedback. First, we would like to clarify that the purpose of designing this position encoding is to enable the model to recognize the relative position of the selected entity node in relation to these patterns. This function is analogous to how the Transformer [A] and ViT [B] architectures handle the relationship between "tokens/patches and their positions."
>
> Specifically, in Association Pattern Sampling (APS) block, the $N$ sampled patterns are assigned to certain node $v$ according to the defined distance. On the one hand, each sampled pattern consists of three entity nodes with $F$-dimension feature embedding, and thus the feature embedding of each pattern is denoted as $\mathbb{R}^{3F}$, which represents the concatenation of the original node feature embeddings, thereby obtaining the feature vector $\mathbf{P}_v \in \mathbb{R}^{N \times(3F)}$ of all the $N$ sampled patterns; on the other hand, each sampled pattern has a distance label defined in Line 154-158 with node $v$. To connect the pattern feature embeddings with the corresponding position information $\mathbf{D}_v \in \mathbb{R}^N$, we provide a position encoding layer $\text{POS}(\cdot)$ which represents a map function from $\mathbb{R}^N$ → $\mathbb{R}^{N \times(3F)}$, and finally integrate $\mathbf{P}_v$ and $\text{POS}(\mathbf{D}_v)$, thereby outputting the enhanced embedding for the next Association Pattern-aware Interaction (API) block.
>
> > W3.3: Explanations for "Bind-relation Feature" and "add" operation in Fig. 2.
>
> A: Thanks for detailed observation. Line 194-199 shows two main functions of the BE module: (1) to reconstruct the missing pairwise feature; (2) to generate hard negative samples. The "Bind-relation Feature" in Fig. 2 actually denotes the above missing pairwise future. Hence, the blue dotted line "delivers" the BE pairwise feature to the APF module, and the icon $\oplus$ "integrates" the two feature embeddings from the BE module and the APF module. Finally, the enhanced feature embedding is fed into the predictor. The above integrating process has been mentioned in Line 219-220, and we will highlight the description with a more separate and clear paragraph.
>
> > Q1: Specific explanation of $\mathcal{L}_{BE}$ (Line 224).
>
> A: Thanks for careful reading and feedback. The $\mathcal{L}_{BE}$ actually denotes the supervised loss of the BE module. The calculation is defined in Line 209, which is equal to the $\mathcal{L}_2$. We will modify $\mathcal{L}_2$ in Line 209  to $\mathcal{L}\_{BE}$.
>
> > Q2: Standard error in Table 1?
>
> A: Thanks for the concern regarding statistical measures. As mentioned in the Appendix B.3.1 and Checklist Q7, each result of these methods in Table 1 is calculated from the average of 5-fold cross-validation experiments with four different scenarios. However, the significant variation in prediction difficulty across different scenarios makes it relatively unreasonable to provide an error bar for all 20 results. Hence, to demonstrate the statistical validity of Pattern-BERP, we have provided all the raw experimental results in Supplementary Materials.
>
>
>
> **References**
>
> [A] Vaswani et al., Attention is all you need. NeurIPS, 2017.
>
> [B] Dosovitskiy et al., An Image is Worth 16x16 Words: Transformers for Image Recognition at Scale. ICLR, 2020.

---

> > ### Comment · Reviewer_UqdA · 2024-08-10
> > **To Authors**
> >
> > I thank the clear and comprehensive response, and my concerns have been mainly addressed especially on the generalization issue, so I raise my final voting to 7. Please provide such discussions in camera-ready.

---

> > > ### Author Response · Authors · 2024-08-10
> > >
> > > We appreciate your acknowledgment of our responses in addressing your concerns. The relevant discussions regarding the extension of variable-length patterns will be elaborated in the final version. Your insights are crucial in enhancing the quality of our work. Thank you once again for the thoughtful review.

---

### Official Review · Reviewer_4Nwa · 2024-07-12

**Soundness:** 3
**Presentation:** 3
**Contribution:** 3
**Rating:** 7
**Confidence:** 2

**Summary:**

This paper presents a novel approach to predicting relationships among biological entities by addressing limitations in current deep learning methods that focus solely on entity-centric information. The authors introduce an association pattern-aware fusion method that integrates association pattern information into entity representation learning. Additionally, they develop a bind-relation module to enhance low-order message passing by considering strong low-order entity associations. Extensive experiments on three biological datasets demonstrate significant improvements compared to advanced baselines. The paper also provides detailed explanations of the interpretability of association patterns, highlighting intrinsic biological mechanisms and potential real-world applications.

**Strengths:**

- The introduction of an association pattern-aware fusion method is innovative. It effectively integrates association pattern information into entity representation learning, which is a novel approach compared to traditional entity-centric methods.
- The method includes several interconnected modules such as association pattern sampling, pattern-aware interaction, and bind-relation enhancement. Extensive experiments on three biological datasets demonstrate significant improvements compared to advanced baselines.
- The method provides detailed explanations of the interpretability of association patterns. This helps in understanding the intrinsic biological mechanisms, making the results more transparent and useful for real-world applications.

**Weaknesses:**

- The proposed model involves multiple modules and several hyperparemeters. Some hyperparameters (such as the number of attention head) are vary on different datasets. The authors may aslo need to add hyperparameter study for loss-balanced coefficient and the threshold for bind-relation prediction probability.
- The approach may not scale well with increasing dataset sizes or more complex biological networks due to its computational requirements and the need for extensive pattern sampling and interaction modeling.
- The proposed model involves multiple components such as hypergraph convolution, pattern-aware fusion, and bind-relation enhancement, which can be computationally intensive. This might limit its applicability for large-scale real-world datasets or in environments with limited computational resources. The author may also need to provide inference speed study with some of the baselines.

**Questions:**

Please see the weaknesses.

**Limitations:**

They provide a limitation section in the appendix.

---

> ### Author Rebuttal · Authors · 2024-08-07
>
> We thank the reviewer for all the comments. We believe we have addressed all the concerns and are happy to follow up in the discussion phase.
>
> > W1.1: Some hyperparameters (such as the number of attention heads) vary on different datasets.
>
> A: We appreciate your valuable feedback. Actually in the machine learning field, performing grid search for hyperparameters is a common practice across different datasets or tasks to achieve optimal results. Therefore, given the significant differences in distribution and association sparsity among datasets, it is entirely reasonable for us to obtain distinct optimal hyperparameter sets, including the number of attention heads mentioned by the reviewer.
>
> > W1.2: More hyperparameter studies for loss-balanced coefficient and the threshold for bind-relation prediction probability.
>
> A: Thanks for your careful reading. We provide additional hyperparameter studies as follows:
>
> * Loss-balanced coefficient $\alpha$. Since the final prediction task involves predicting the associations among entity A, B, and C, we intuitively consider the two tasks of A→B and B→C to be of equal importance and thus the $\alpha$ is set to 0.5 by default. Here, we attempt to adjust $\alpha$ from 0.1 to 0.9, and the results are shown below (the hits@1 with default $\alpha$=0.5 is 93.94). The hits@1 performance of the default setting surpasses other ablation settings.
> | coefficient $\alpha$ | hits@1 | coefficient $\alpha$ | hits@1 |
> | -------------------- | ------ | -------------------- | ------ |
> | 0.1                  | 92.91  | 0.6                  | 92.79  |
> | 0.2                  | 93.87  | 0.7                  | 93.25  |
> | 0.3                  | 93.86  | 0.8                  | 93.66  |
> | 0.4                  | 93.32  | 0.9                  | 93.23  |
>
> * Threshold $\gamma$ for bind-relation prediction probability. Bind-relation prediction is fundamentally a binary classification task, so setting the threshold to 0.5 by default is quite common. Here, we also attempt to adjust the threshold $\gamma$ from 0.1 to 0.9, and the results are shown below (the hits@1 with default $\gamma$=0.5 is 93.94). The hits@1 performance of the default setting surpasses other ablation settings. It is noteworthy that when $\gamma$=0.1, the generation of negative samples is extremely hard as defined in Eq. 9, and thus the BE module cannot provide valid hard samples.
> | threshold $\gamma$ | hits@1           | threshold $\gamma$ | hits@1 |
> | ------------------ | ---------------- | ------------------ | ------ |
> | 0.1| No Valid Samples | 0.6| 92.17  |
> | 0.2| 91.88 | 0.7| 93.67  |
> | 0.3| 92.91 | 0.8| 93.88  |
> | 0.4| 93.48 | 0.9 | 93.81  |
>
> > W2: Computation complexity explosion for huger dataset or more complex biological networks.
>
> A: Thanks for your valuable insight. As discussed in Q1 by reviewer ugjG, the total complexity of the entire graph is denoted as $ \mathcal{O}\left(|\mathcal{V}| \cdot (N^2 \cdot d + N \cdot d^2 + N \cdot d \cdot f) \right)$, where $|\mathcal{V}|$ represents the number of entity nodes, $N$ represents the number of sampled patterns,  the dimensions of all input pattern features and hidden features in MHA layer are $d$ and the hidden features in FFN layer is $f$. It is evident that computational complexity is primarily determined by the number of entity nodes $|\mathcal{V}|$ and the number of sampled patterns $N$.
>
> Hence, when encountering larger datasets or more complex biological networks, we can employ the following two strategies to avoid a computation complexity explosion: (1) Graph Sampling. Processing large graph data requires significant computational resources. By sampling smaller subgraphs [A], computational resource consumption can be significantly reduced, improving the speed and efficiency of algorithms; (2) High-confidence Pattern Selection. As shown in Fig. 4 (g)-(i), reducing $N$ from 100 to 5 across three datasets slightly decreases performance but still surpasses these baselines. Thus, adjusting the sampling quantity is acceptable to effectively reduce time complexity.
>
> Furthermore, to test on larger and diverse datasets, our approach is refined to capture variable-length patterns and applied to large-scale molecular property datasets. As shown in Table R2 (response PDF), the association patterns significantly enhance atom (analogous to entity nodes in this study) representations and improve prediction performance.
>
> > W3.1: Limitation of applying in large-scale real-world datasets or environments with limited computational resources.
>
> A: Thanks for your further concern regarding the real-world scenarios. As discussed in W2, the computation complexity can be controlled through some reasonable strategies, such as graph sampling and pattern selection. Moreover, substantial computational resources are typically deployed on central servers in the field of bioinformatics, unlike the field of computer vision, which often requires deployment on resource-constrained devices.
>
> > W3.2: Inference speed study compared to baselines.
>
> A: Thanks for highlighting this aspect. We provide the inference speed comparison with other methods as follows and present the inference time of each 100 samples with milliseconds. Compared to the baseline methods, the difference in inference time for our method is constant and even faster than several baselines.
>
> | Method | DMD (ms) | DDC (ms) | DPA (ms) |
> | ---------------- | -------- | -------- | -------- |
> | RF | 0.36|0.66| 0.57 |
> | MLP | 0.18| 0.21 | 0.26 |
> | CP| 8.98| 9.19 | 10.25|
> | Tucker| 7.92 | 8.84 | 7.96|
> | CoSTCo| 0.75 | 1.51| 3.15|
> | GCN | 0.80 | 0.99| 1.95|
> | GraphSAGE | 0.81 | 0.95| 1.92|
> | GAT | 1.18 | 1.61 | 3.11 |
> | GIN | 0.72| 1.14 | 1.86|
> | DHNE  | 0.29 | 0.29 | 0.39|
> | HyperSAGNN | 0.63 | 0.69| 0.56|
> | HGSynergy | 0.64 | 0.89| 1.55 |
> | MCHNN (CPU-only) | 8.65 | 9.54 | 13.72 |
> | Pattern-BERP | 1.28 | 1.24| 2.96|
>
>
>
> **References**
>
> [A] Hu and Lau, A survey and taxonomy of graph sampling. arXiv, 2013.

---

> > ### Comment · Reviewer_4Nwa · 2024-08-09
> > **To Authors**
> >
> > Thanks for your responses. I think my rating is reasonable and fair.

---

> > > ### Author Response · Authors · 2024-08-10
> > >
> > > Thanks for your efforts and the considerable recognition of our paper. The quality of the manuscript will be greatly improved in accordance with your valuable comments.

---

### Official Review · Reviewer_m3DY · 2024-07-12

**Soundness:** 3
**Presentation:** 2
**Contribution:** 2
**Rating:** 6
**Confidence:** 3

**Summary:**

The author proposed a deep learning method, termed the Pattern BERP method, designed to elucidate the potential associations among triple-wise biological entities. This method innovativly incorporates association pattern information into the entity representation learning. Moreover, it employs two additional bipartite graphs to enrich the binary relation information extracted from triple entities. Evaluated on three biological datasets, the proposed method demonstrates superior performance compared to non-graph, graph-based, and hypergraph-based methodologies. This work is noteworthy not only for its impressive predictive accuracy but also for its potential applicability in real-world scenarios.

**Strengths:**

1.	The concept of k-hop neighbors on a hypergraph has been defined, incorporating pattern-similar triples information as neighbors into each hypernode.

2.	The information of triples has been decomposed into pairwise relationships, and these features have been injected into the backbone network.

3.	A parameter-based negative sampling technique has been introduced, which is more efficient compared to random sampling.

**Weaknesses:**

1. The interpretation provided for the model is not entirely persuasive. The analysis of the relationship between peptide-based micro-molecular drugs and small-molecular chemical drugs is unconventional and lacks commonality.

2. While the concept of association pattern-aware fusion is highly innovative and appropriately highlighted as the main focus of the paper, its contribution for model performance seems not to be impressive, especially in Table 5 and Table 6.

3. The model is overly complex, with multiple stages of tasks and losses, which may lead to difficulties in optimization.

4. The role of the triple information has been simplified to a single chain pattern from A to B to C.

**Questions:**

1. The abbreviation "APF" (association pattern fusion) is not defined in the text at line 219. It is essential to provide the full name when first introduced for clarity.

2. In equation (11), the LBE is not updated every epoch, which contradicts the description in the text.

3. The manuscript features two bipartite graphs in Figure 2, yet it remains unclear whether these graph features are integrated with hypergraph features. Additionally, the role of the blue dotted line arrow, which is not discussed in the text, needs clarification to enhance understanding of the graphical representations.

4. Although the association pattern-aware fusion is novel and central to the paper's title, its contribution for model performance seems not to be impressive. Specifically, while Table 2 shows significant impairment in performance on the DDC dataset without the association pattern interaction (API), Tables 3 and 4 for the DMD and DPA datasets suggest that the Binding-relation Feature Reconstruction (BFR) for bipartite graphs predominantly enhances performance. It may be misleading to attribute the main contribution of the work solely to APF given these findings.

5. The model interpretation in Figure 3 lacks persuasiveness, as Drug #52, a peptide with numerous sub-group structures, is not ideal for visualization comparison with other compound drugs. Furthermore, in equation (3), the attention mechanism is applied across different triple entities, not just one. Analyzing only the drugs without considering the other two entities in the figures is insufficient. At minimum, the roles of the other two entities should be elucidated to provide a more comprehensive analysis.

6. Equation 2 describes the generation method of the distance matrix D and the distance token 𝐷𝑣. How is the specific position embedding done on line 167? Is the initial pattern 𝑃 directly concatenated from the three nodes?

7. In the bind-relation enhancement, why is only the bipartite graph between AB and BC considered? Why not consider AC? Not all multi-entity relationships have a clear chain pattern from A to B to C.

8. Following up on question 7, how are relationships defined between two entities in Equations 7 and 8?

**Limitations:**

The model lacks practical application scenarios. It primarily learns from the interaction network of existing entities such as drugs, diseases, and symptoms. Such methods generally struggle to predict the relationships between a completely new entity and the existing entities in the network. Moreover, the relationships within the network are already well-known. The more important task is to uncover new and potential relationships. The comprehensiveness and accuracy of the existing relationships are merely the tasks designed by the model itself.

---

> ### Author Rebuttal · Authors · 2024-08-07
>
> We thank the reviewer for all the comments. We believe we have addressed all the concerns and are happy to follow up in the discussion phase.
>
> > W1: The provided interpretation study lacks absolute persuasiveness.
>
> A: Thanks for your insight. There are two additional interpretation cases in Fig. R1 (response PDF), which shows the common patterns closely linked to the original drug node reflect the mechanism that acts on the cell wall and envelope and thus influences microbe physiological activities, thus treating diseases. In contrast, the low-contribution pattern causes cell death by inhibiting protein synthesis. We will provide detailed text descriptions for interpretation cases in the future.
>
> > W2: The contribution of APF seems not to be impressive in Table 5&6. Which module contributes more to three different datasets?
>
> A: Thanks for your valuable feedback. The situation arises due to significant differences in the distribution and sparsity of these three datasets (see Appendix Table 3). Table 2 shows that APF's contribution stands out in the DDC. However, in the relatively sparse DMD and extremely sparse DPA, there are fewer high-confidence patterns associated with entity nodes (some nodes link to only one pattern), reducing the APF module's effectiveness. In the future, as biological association datasets grow larger and denser, APF's pattern exploration will yield more expressive representations, making its contribution more impressive.
>
> > W3: Overly complex model architecture with multiple stages of tasks and losses.
>
> A: Thanks for concern. The core components are two complementary modules: Association Pattern-aware Fusion (APF), Bind-relation Enhancement (BE) with their corresponding losses. Due to the limited response characters, please refer to W1 of reviewer UqdA for details.
>
> > W4: The triple information is simplified to a single chain pattern from A to B to C.
>
> A: Thanks for pointing this out. In this study, the triplet relationship we investigate is defined as the A→B→C pattern due to dataset limitations, making it challenging to collect substantial longer physiological pathway data. As noted in the Limitation section, we plan to extend to longer pattern pathways and have already obtained some preliminary results.
>
> > Q1: Full name of APF when first introduced.
>
> A: Thanks for careful reading. The abbreviation APF denotes Assocaition Pattern-aware Fusion, which will be introduced in Line 149.
>
> > Q2: Ambiguity for total loss function in Section 4.4.
>
> A: Thanks for carefulness. The total function is revised by the rule: BE module trains for the first 4 epochs of each 5-epoch cycle; APF module trains for the final epoch. Please refer to W1 of reviewer UqdA for details.
>
> > Q3: Unclear description for bipartite graph feature in Fig. 2 and its caption.
>
> A: Thanks for pointing it out. The bipartite graph features are integrated with hypergraph features, represented by the blue dotted line arrow. Please refer to W3.3 of reviewer UqdA for details.
>
> >  Q4: The peptide is not ideal for visualization comparison with other compound drugs.
>
> A: We understand your concern. Additional cases in Fig. R1 (response PDF) are both discussed with small molecule drugs.
>
> > Q5: The roles of the other two entities should be elucidated in the interpretation study.
>
> A: Thanks for highlighting this aspect. Due to insufficient dataset information, we cannot obtain identifiers for Microbe and Disease entities, which are directly processed into feature vectors in the original DMD dataset [17]. In the future, we will focus on intrinsic interpretation using more comprehensive datasets.
>
> > Q6: Details about how the position embedding works (line 167). Is the initial pattern $P$ directly concatenated from the three nodes?
>
> A: Thanks for pointing it out. The purpose of designing this position encoding is to enable the model to recognize the relative position of the selected entity node in relation to these patterns. Please refer to W3.2 of reviewer UqdA for details.
>
> In this work, each sampled pattern consists of three different types of entity nodes representing a reaction pathway. These patterns may contain common knowledge (see Fig. 2), aiding in learning more expressive representations for entity nodes. Additionally, in the Limitation section, we propose exploring variable-length patterns to identify more representative pathways.
>
> > Q7: Why not consider A→C in the bind-relation enhancement?
>
> A: Thanks for your concern. Our focus is on the triplet relationships like Drug→Protein→ADR discussed in the paper. Typically, a drug interacts with a target protein, triggering signals that cause various activities, some manifesting as ADRs like vomiting or diarrhea. Therefore, we did not consider a direct A→C relationship, as there is no connection in this context. However, we acknowledge your perspective on broader multi-pathway relationships where a direct A→C could exist. We will consider it for future tasks.
>
> > Q8: The defined relationships between two entities in Equations 7 and 8.
>
> A: Thanks for highlighting this aspect. In this paper, A→B and B→C have clear biological associations, reflected in the data through labels. However, the A→C relationship is determined by modeling new negative samples A→B→C (Bottom right of Fig. 2, Line 211-217).
>
> > L: The model struggle to predict the relationships between a completely new entity and the existing entities in the network, thus lacking practical application scenarios.
>
> A: Thanks for guidance on future direction. Our core goal is to uncover common association patterns between biological entities, which remain unchanged with the addition of new nodes. Conversely, by linking new nodes to existing ones through common patterns, we aim to uncover new potential relationships. Of course, we highly appreciate your viewpoint. The current model design indeed lacks the ability to introduce new entity nodes. We will improve it in the future to facilitate the practical application.

---

> > ### Comment · Reviewer_m3DY · 2024-08-14
> >
> > Thanks for detailed response. I have raised the score.

---

> > > ### Author Response · Authors · 2024-08-14
> > >
> > > We appreciate your effort on our paper and the recognition of our responses. Your unique insights and detailed review have significantly enriched our work, allowing us to further refine this manuscript. Thank you once again for your valuable feedback.

---

### Official Review · Reviewer_ugjG · 2024-07-12

**Soundness:** 3
**Presentation:** 3
**Contribution:** 3
**Rating:** 7
**Confidence:** 3

**Summary:**

This work presents a novel approach - Pattern-BERP - for the prediction of biological entity relationships, it utilises entity association patterns as opposed to a lot of existing research that focuses primarily on entity-centric information mapping and aggregation.

The evaluation is done on three different biological datasets (DMD, DDC, DPA), and compared against a range of different approaches including non-graph, graph-based and hypergraph-based ones.

The results show state-of-the-art performance across the board (an increase from 2 to 10 points over the next best method).

The authors also do an ablation study, showing the importance of each component and demonstrating the relative improvement.

**Strengths:**

A novel idea presented in an easy-to-understand way. Manuscript is nicely written and clear, with a broad impact. An extensive evaluation concerning methods, state-of-the-art results and a detailed ablation study.

**Weaknesses:**

This is a very general method but it was tested on only one dataset type and relatively small datasets. Would be great to see it expanded in the future.

**Questions:**

Similar to the limitations field, I'd be very interested in an explanation of how the computation complexity of Pattern-BERP scales with the number of entities and associations.

Is it possible to provide a bit more theory behind the success of your method compared to others? The justification for the association pattern-aware fusion to me seems mostly intuitive rather than theoretical.

Is there a way to extend the method to capture variable-length pathways?

**Limitations:**

It would be great to see a longer discussion on limitations: my primary concerns are complexity explosion and very large datasets. Would be great to see a bit of a theoretical analysis (or even empirical) on how it scales for millions of entities or associations (which are not rare in the domain).

The interpretability section focuses on only drug #52, it would be great to see that expanded, as making conclusions from one example is bit of a stretch.

Real-world applications might be a bit of an overstatement, while it is possible, nothing was really tested or proven.

---

> ### Author Rebuttal · Authors · 2024-08-07
>
> We thank the reviewer for all the comments. We believe we have addressed all the concerns and are happy to follow up in the discussion phase.
>
> > W: Possibility to test on more dataset types and relatively huge datasets?
>
> A: We appreciate your insight. Firstly, we clarify that the limited associations in this study result from current research constraints, despite large Cartesian products in Appendix Table 3. Previous works [15-17,19] also used similar dataset types. To test on larger, diverse datasets, our approach is refined to capture variable-length patterns and applied to large-scale molecular property datasets. As shown in Table R2 (response PDF), the association patterns significantly enhance atom (analogous to entity nodes in this study) representations and improve prediction performance.
>
> > Q1: How the computation complexity of Pattern-BERP scales with the number of entities and associations?
>
> A: Thanks for your concerns. The most critical module is undoubtedly the APF, which aggregates multiple patterns across different entity nodes. This module includes Multi-Head Attention (MHA) and FeedForward Network (FFN). For an entity node with $N$ sampled patterns from the APS module, the input and hidden features in the MHA layer are of dimension $d$, and hidden features in the FFN layer are $f$, and the query, key, and value matrices are derived from the same input sequence and share length $N$. In APF, the primary operations include scaled dot-product attention, multiplication of attention weights and values, MHA linear transformation, and FFN linear projection. The time complexity is expressed as $\mathcal{O}(N^2 \cdot d + N \cdot d^2 + N \cdot d \cdot f)$. Hence, for the entire graph, the total complexity is $\mathcal{O}\left(|\mathcal{V}| \cdot (N^2 \cdot d + N \cdot d^2 + N \cdot d \cdot f)\right)$, where $|\mathcal{V}|$ is the number of entity nodes. Moreover, Table R1 (response PDF) shows the GPU usage of Pattern-BERP scaling with the number of entities and patterns.
>
> > Q2: Theory behind the success of the method.
>
> A: Thanks for highlighting this point. Pattern-BERP is regarded as an empirical method. It extracts common patterns or rules from large bio-associated datasets, similar to k-means clustering. Given a triplet dataset $\mathcal{D}=\\{\mathbf{x}\_1,\mathbf{x}\_2,\ldots,\mathbf{x}\_n\\}$ and $k$ common patterns, our goal is to minimize the total distance of samples to their respective pattern centers within the same pattern. First, initialize the pattern centers $\mathbf{m}\_1,\mathbf{m}\_2,\ldots,\mathbf{m}\_k$. Next, assign each sample to the nearest pattern center by computing $j=\arg\min_k \\|\mathbf{x}\_i-\mathbf{m}\_k\\|$. Then, update the pattern centers using $\mathbf{m}\_j=\frac{1}{|C_j|}\sum_{\mathbf{x}\_i \in C_j}\mathbf{x}\_i$, where $C_j$ is the set of samples assigned to pattern $j$. Iterate these assignment and update steps until the pattern centers converge. The objective function to minimize is $J=\sum_{j=1}^{k}\sum_{\mathbf{x}\_i \in C_j} \\|\mathbf{x}\_i - \mathbf{m}\_j\\|^2$, and by minimizing $J$, Pattern-BERP extracts representative biological association patterns effectively. In the future, additional corresponding theoretical analysis will be provided in detail.
>
> > Q3: Method extension to capture variable-length pathways.
>
> A: Thanks for constructive advice. As discussed this limitation in Appendix Section C, we strived to extend the method to capture variable-length patterns after the paper submission. Since now, the proposed method has been optimized with variable-length settings and thus applied on the molecular property prediction tasks to enhance the atom (in line with entity in this paper) representation in the molecular graph. Table R2 (response PDF) demonstrates the effectiveness of variable-length association patterns.
>
> > L1: Discussion on very large datasets and complexity explosion.
>
> A:  We understand your primary concern. Regarding computation complexity, the discussion in Q1 indicates that the decisive factors are the number of entity nodes $|\mathcal{V}|$ and the number of sampled patterns $N$.
>
> Hence, when encountering larger datasets or more complex biological networks, we can employ the following two strategies to avoid a computation complexity explosion: (1) Graph Sampling. By sampling smaller subgraphs [A], computational resource consumption reduces significantly, boosting algorithm speed and efficiency; (2) High-confidence Pattern Selection. As shown in Appendix Fig. 4 (g)-(i), reducing $N$ from 100 to 5 across three datasets slightly decreases performance but still surpasses these baselines. Thus, adjusting the sampling quantity is acceptable to effectively reduce time complexity.
>
> Moreover, Table R1 (response PDF) shows the GPU usage varying from the number of sampled pattern across three datasets with significant differences in the number of entities and associations. The inference time compared to baselines can be found in W3.2 of reviewer 4Nwa.
>
> > L2: Interpretability analysis of more cases.
>
> A: Thanks for your insight. There are two additional interpretation cases in Fig. R1 (response PDF), which shows the common patterns closely linked to the original drug node reflect the mechanism that acts on the cell wall and envelope and thus influences microbe physiological activities, thus treating diseases. In contrast, the low-contribution pattern causes cell death by inhibiting protein synthesis.
>
> > L3: Restate real-world applications of the proposed method.
>
> A: Thanks for pointing this out. Actually in Appendix Section C, we would have intended to discuss the possibility of applying our proposed APF-BERP in real-world scenarios and the corresponding limitations. Due to limited resource and time, we cannot indeed validate it with enough real tests or proofs. We will restate the confusing description about the real-world applications.
>
>
>
> **References**
>
> [A] Hu and Lau, A survey and taxonomy of graph sampling. arXiv, 2013.

---

> > ### Comment · Reviewer_ugjG · 2024-08-13
> > **All clear**
> >
> > Thank you for the answers, all is clear.

---

> > > ### Author Response · Authors · 2024-08-13
> > >
> > > Thank you for your time and effort on our paper. We are delighted to hear that our responses are clear and satisfactory. Your thorough review and constructive feedback have significantly contributed to the refinement of our manuscript.

---

### Author Rebuttal · Authors · 2024-08-07

We thank all reviewers for the time spent reviewing the paper and recognizing the **significance** ("with a broad impact" – ugjG, "noteworthy" & "impressive predictive accuracy" & "more efficient" – m3DY, "significant improvements" – 4Nwa), **novelty** ("novel idea" – ugjG, "highly innovative" – m3DY, "innovative" & "a novel approach" – 4Nwa, "novel and explainable" & "reasonable" – UqdA) of our research direction, the **quality of the presentation** ("nicely written and clear" & "presented in an easy-to-understand way" – ugjG, "clear and detailed" & "comprehensively discusses" – UqdA), and **applicability** ("potential applicability in real-world scenarios" – m3DY, "more transparent and useful for real-world applications" – 4Nwa).

We have endeavored to consider the feedback as comprehensively as possible, leading to a revision process that significantly honed the paper. We have addressed every point in our responses and are happy to follow up on any aspect during the discussion phase. Specifically, we have tackled stated weaknesses (W), questions (Q), and limitations (L) with detailed answers (A). Additionally, we have included a single-page PDF containing extra referenced figures and tables to further clarify our points.

Furthermore, we have strived to balance making all necessary changes within the given space constraints. However, due to the response length limitations, our answers to some common but important problems that require detailed explanation may refer to the response provided to another reviewer, indicated by the statement "please refer to the response for another reviewer." We sincerely apologize for any inconvenience this may cause.

Finally, we would like to express our appreciation once again for the reviewers' constructive comments and careful reading, which undoubtedly lead to enhancing the quality of our work.

---

### Decision · Program_Chairs · 2024-09-25

**Decision:**

Accept (poster)

**Comment:**

This work seeks to predict relationships between biological entities,  via an association pattern-aware fusion method that integrates association pattern information into entity representation learning.  It is a solid piece of work - the    association pattern-aware fusion method is creative.